# Modeling CMAQ dry deposition treatment over Western Pacific: A distinct characteristic of mineral dust and anthropogenic aerosol

Steven Soon-Kai Kong[1], Joshua S. Fu[2], Neng-Huei Lin[1, 3, *], Guey-Rong Sheu[1, 3, *], Wei-Syun Huang[1]

[1] Department of Atmospheric Sciences, National Central University, Taoyuan, 32001, Taiwan
[2] Department of Civil and Environmental Engineering, the University of Tennessee Knoxville, TN37996, USA
[3] Center for Environmental Monitoring and Technology, National Central University, Taoyuan, 32001, Taiwan

*Correspondence to*: Neng-Huei Lin (nhlin@cc.ncu.edu.tw) and Guey-Rong Sheu (grsheu@atm.ncu.edu.tw)

**Abstract.** Dry deposition plays a vital role in the aerosol removal process from the atmosphere. However, the chemical transport model (CTM) is sensitive to the dry deposition parameterization and yet remains to be determined due to the limited particle deposition measurement. By utilizing the CMAQv5.4 with the refined dust emission treatment, the East Asian dust (EAD) simulation during January 2023 and Spring 2021 was constructed to evaluate the performance of dry deposition parameterizations, namely S22, E20, and P22. The result showed that the dry deposition parameterization could significantly impact the CMAQ dust concentration in the air. By implementing the E20 dry deposition scheme, the CMAQ simulation performance of the surface $PM_{10}$ has been considerably improved with the NMB of -41.9 %, as compared to the dry deposition proposed by S22 (-47.01 %) and P22 (-53.90 %). The modeled $PM_{10}$ pattern by E20 at the upper level (700 hPa) was mostly consistent with the observed $PM_{10}$ at the Lulin Atmospheric Background Station (LABS; 23.47° N, 120.87° E; 2862 m a.s.l.) where is a typical background site at Western Pacific, particularly in capturing the peak value. The correlations (R) at the high-altitude were well performed for E20 by 0.55, as compared to S22 (0.54) and P22 (0.46). Moreover, E20 improved the simulated $PM_{10}$ concentrations and aerosol optical depth (AOD) value over the Asian Continental during the multiple dust episodes in spring 2021, by NMB of -25.43 % and -26.19 %, respectively. The noticeable deduction of the coarse mode particle's deposition velocity ($V_d$) was responsible for reducing the $PM_{10}$ simulation underestimation. On 22-31 January 2023, the *in-situ* measurement of the upper level observed the possibility of natural dust and anthropogenic aerosol. This is consistent with the CMAQ, which shows that both aerosol types displayed a clear "long dust-black

carbon belt" along the 15°N. It is revealed that the increase of the surface resistivity ($R_b$) leads to a
significant increase in dust mass concentration but a minor increase in black carbon (BC). We proposed
implementing the E20 dry deposition approach, particularly in $PM_{10}$ simulation, narrowing the
uncertainty of the CMAQ dust emission treatment.

## 1 Introduction

The chemical transport model (CTM) is a powerful tool for comprehending air pollution, encompassing
emission, transport, radiative impact, and removal mechanisms at various grid scales. Among
these, particle dry deposition is a crucial aerosol removal process and an important sink for particles in
the model. The derivation of the dry deposition is based on the resistance framework and
electrical analogue, but its implementation can vary across models (Wesley, 1989; Giardina and Buffa,
2018; Gaydos et al., 2007; Khan and Perlinger, 2017; Shu et al., 2017). A key challenge in dry deposition
simulation is the scarcity of measurement data for model verification, underscoring the necessity for
further research to enhance the accuracy of air quality modeling.
An immense range of dry deposition parameterization has been implanted in the model. The
deposition mechanism by Slinn (1982) includes the deposition process such as turbulent transfer,
Brownian diffusion, impaction, interception, gravitational settling, and particle rebound, where the
particle grows under humid conditions. Zhang et al. (2001) suggested the dry deposition scheme is
sensitive to land use category and several parameters. For instance, due to the particle growth, the
deposition velocity ($V_d$) over the ocean is much higher than on another land surface, as the $V_d$ increased
rapidly with the increase of particle size. Some CTMs using Zhang et al. (2001) parameterization still
underestimated the global $PM_{2.5}$ concentration. The latest dry deposition scheme revision by Emerson et
al. (2020) based on the flux measurement of grassland and pine forest has reduced the uncertainty,
marking a significant step forward in our quest for more accurate air quality modeling.
An updated deposition scheme that reduces the dependence of the deposition velocity on the aerosol
mode width has been proposed (Shu et al., 2022). Indeed, the approach suggested that vegetation
dependence increased the $V_d$ for submicrons and decreased for large particles by 37 % and -66 %,

respectively. It also reduced the functional biases by 56-97 % for vegetated land-use type and equivalence performance over the water. Moreover, adding the second inertial impaction term for microscale obstacles such as leaf hairs, microscale ridges, and needle leaf edge effects managed to increase the mass dry deposition of the accumulation mode aerosols in the model (Pleim et al., 2022). These modifications reduced the average $PM_{2.5}$ in the atmosphere during July 2018 over the contiguous United States.

With a plethora of deposition approaches in use, it becomes paramount to comprehend their impact on model performance in predicting aerosol behaviour. The surface fine particle concentrations can vary up to 5-15 %, and the particle dry deposition has more than 200 % discrepancy due to the different dry deposition schemes. (Saylor et al., 2019). A comprehensive evaluation of five different parameterizations has been conducted, with the simplest and most effective deposition mechanism suggested for the CTM (Khan and Perlinger, 2017). However, the model's reliance on meteorological factors such as frictional velocity, relative humidity, rainfall, or wind speed, which can significantly influence the model's accuracy, remains a challenge (Kong et al., 2021).

Besides the model bias on $PM_{2.5}$, the simulation of $PM_{10}$ has been underestimated due to the uncertainty of the deposition mechanism, particularly over the western Pacific (Kong et al., 2021). The $V_d$ is overestimated for coarse particles, where the dry deposition velocity is too high for coarse particles when the frictional velocity is large, which is why the surface $PM_{10}$ concentration is underestimated (Ryu and Min, 2022). The model performance of $PM_{10}$ simulation that is widely influenced by the dust treatment embedded within CMAQ has been revised (Dong et al., 2016; Liu et al., 2021; Kong et al., 2021, 2024) and are found to effectively simulate the $PM_{10}$ over the western Pacific region such as Taiwan. However, the issue regarding the deposition algorithm's impact on the model performance at the corresponding region needs to be discussed. The present research intends to evaluate the CMAQ model performance due to the different deposition schemes on aerosols in the Taiwan region.

The model performance in Taiwan is paramount in our study, as the area is equipped with a substantial number of well-maintained surface observation sites, providing comprehensive coverage. The LABS station in the high-altitude subtropical western North Pacific region serves as the sole background station

for monitoring transboundary pollutants. This station is crucial in our research as it provides unique data on the long-range transport of pollutants, further underscoring the relevance of our study.

The transboundary pollutants mechanisms have been widely discussed through LABS measurements, cooperating with the backward trajectory, reanalysis dataset, and modeling approach. Previous research reveals that LABS pollutants could be associated with severe fire emissions from northern Peninsular Southeast Asia (Huang et al., 2020; Ooi et al., 2021) and Indonesia (Ravindra Babu et al., 2023). Moreover, the intense wind speed in northwest China could transport the mineral dust through the surface and high-altitude layer detected at LABS (Kong et al., 2021; Kong et al., 2022). Additionally, the transport process of East Asian haze due to the cold surge from the Asian Continental industrial region towards Taiwan has been widely discussed (Chuang et al., 2020). Instead of pure aerosol, the coexistence of dust and biomass burning over Taiwan, a condition discovered in previous research, has significant implications for the regional climate (Dong et al., 2018; Dong et al., 2019). However, the high-altitude synoptic pattern associated with the coexistence between natural dust and anthropogenic pollutants remains unknown due to a lack of observations at the upper layers.

This study used the chemical transport model to investigate the long-range transport of East Asian dust (EAD) that occurred on 22-31 January 2023 and 12 March-20 April 2021. Due to the limitation of the dust model, the CMAQ version 5.4, embedded with three types of dry deposition schemes, was implemented to justify the effectiveness of improving our latest refined dust model (Kong et al., 2024). The dry deposition scheme proposed by Shu et al. (2022) has reduced certain model bias as compared to the base scheme. However, the revised scheme response to the natural phenomenon such as wind-blown dust has not being tested. In the other way, the number of concentrations of the large size particle has been decreased over land, and increased over ocean area globally by the adjusted collective coefficiency proposed by Emerson et al. (2020). Pleim et al. (2022) has included the consideration of white cap effect which dependent on wind speed and sea surface temperature into the dry deposition scheme. Hence, the response of the CMAQ dust model under the newly developed dry deposition schemes are worth investigating in reducing the model uncertainty.

LABS detected the recent transboundary episode in January 2023 as a mixing aerosol type (see Section 3.1), which has not been widely discussed, and the multiple dust storm episodes mentioned by Kong et al. (2024) provide an opportunity to model the EAD over the downwind region. Recognizing the significant transboundary events detected through Taiwan's observations, the improvement of the CMAQ dust model by the dry deposition schemes, and its application in characterizing the transport mechanism can be vital. The paper is organized as follows. The model setup and ancillary datasets are discussed in Sect. 2. The results and discussion are presented in Sect. 3, followed by the conclusions in Sect. 4.

## 2 Data and Methodology

### 2.1 Dust emission treatment

Before delving into the details, it's important to understand the process of dust transport. Dust is primarily transported by wind through a process known as sandblasting (Kok et al., 2012). For dust to be uplifted, the horizontal wind speed must exceed a certain threshold frictional velocity ($u_{*,t}$), which is estimated by the model as follows:

$$u_{*,t} = u_{*,to} f_m f_r \tag{1}$$

Where $u_{*,to}$ is the ideal threshold friction velocity, while $f_m$ and $f_r$ are the correction factors of soil moisture and surface roughness, respectively.

Through a collaborative effort, the windspeed, soil texture, soil moisture, and surface roughness length derived from field and laboratory studies have been integrated into the windblown dust treatment, which is now a part of the Community Multiscale Air Quality (CMAQ) modeling system (Foroutan et al., 2017). This model, developed and evaluated over the continental United States, has also been extended to the East Asia region (Dong et al., 2016; Liu et al., 2021; Kong et al., 2021, 2024). Kong et al. (2024) have proposed further improvements, including the integration of the revised soil moisture fraction, dust emission speciation profile, and bulk soil density, to enhance the representation of the Asian dust simulation. This ongoing collaboration is crucial for the continuous improvement of our understanding and management of dust emissions.

## 2.2 Particle dry deposition schemes

Particle dry deposition is a complex process relating to the deposition velocity, particle size, source and composition, land use surface, and meteorological condition. Generally, the flux of the particle mass through the surface boundary layer is estimated as:

$$F = C \times V_d \tag{2}$$

where F is the deposition flux, C is the particle concentration at the surface layer, and $V_d$ is the deposition velocity.

The difference in the particle concentration and deposition prediction among the various atmospheric chemistry models was probably due to the algorithm of the dry deposition particle. The algorithm describing particle deposition velocity as a function of particle size in almost all current air quality model systems is descended from Slinn (1982). The particle deposition according to vegetative canopies formulated the deposition velocity as:

$$V_d = V_g + \frac{1}{R_a + R_b} \tag{3}$$

where $V_g$ is the gravitation settling velocity, $R_a$ is the resistivity aerodynamic, $R_b$ is the surface resistivity, also known as quasi-laminar sub-layer resistivity in STAGE. The $V_g$ is calculated according to Stokes's Law as:

$$V_g = \frac{p_p \, D_p^2 \, g C_c}{18\eta} \tag{4}$$

where, $p_p$ is the density of the particle; $D_p$ is the diameter of the particle; g is gravitational acceleration; $C_c$ is the Cunningham correction factor for small particles; and, $\eta$ is the dynamic viscosity of air.

CMAQ is embedded with M3Dry dry deposition calculation that implements the scheme of Pleim and Ran (2011), which is based on Slinn (1982). As noted by Pleim and Ran (2011), chemical surface flux modeling has become an essential process in the air quality model. For instance, the linkages of

ambient concentration levels to the deposition of $SO_x$ and $NO_x$. Moreover, Surface Tiled Aerosol and Gaseous Exchange (STAGE) deposition has been implemented within the CMAQv5.3, where estimated fluxes from sub-grid cell fractional land-use values, aggregate the fluxes to the model grid cell and unifies the bidirectional and unidirectional deposition schemes using the resistance framework (Massad et al., 2010; Nemitz et al., 2001). The updated STAGE version in CMAQv5.4 could aggregate the grid-scale values that match the grid-scale values from most kinds of Land Surface Model of WRF (Hogrefe et al., 2023). Since the present study is primary focused on the impact of dry deposition scheme on CMAQ dust modeling, the simulations with the STAGE module are the mandatory concern (Table 1).

## 2.3 CMAQ model design

This study applied WRF v4.0 for the meteorological field parameters and CMAQv5.4 to simulate the transboundary East Asian dust episodes on 22-31 January 2023, and the multiple dust storm episodes during 12 March-20 April 2021. The modeling domain was set up to cover the Taklamakan and Gobi Desert, with a resolution of 45 km, and nested towards Taiwan at a resolution of 15 km (d02) and 5 km (d03) (Fig .1, Table 2). Also, as Taiwan is influenced by biomass burning, the domain covers up to the peninsular Southeast Asia (PSEA), which will be carried out in the future (Ooi et al., 2021). The model consisted of 40 vertical layers, with eight layers below~1 KM altitude, 13 layers below ~3 KM altitude, and 27 layers covering the upper layer to ~21 KM. The model's initial and lateral boundary conditions were constructed using the National Centers for Environmental Prediction (NCEP) Final Analyses (FNL) reanalysis dataset on a 0.5◦ ×0.5◦ grid. The data assimilation was conducted by grid nudging in all the domains. The CB06 gas-phase chemical mechanism and the AERO7 aerosol module model were implemented in CMAQ for the present study.

The anthropogenic emission inventories in East Asia, crucial for our research, were obtained from the MICS-Asia (Model Inter-Comparison Study for Asia) Phase III emission inventory (Li et al., 2017). The emissions of $SO_2$, NOx, NMVOC, $NH_3$, CO, $PM_{10}$, $PM_{2.5}$, BC, OC and $CO_2$ has been meticulously modified, taking into account of the relative changes in China's anthropogenic emissions between 2010 and 2017 (Zheng et al., 2018). Additionally, the modified emission of $NO_2$ was adjusted further by the satellite imagery OMI-$NO_2$ in January 2023 (Huang et al., 2021). Biogenic emissions for Taiwan were

prepared by the Biogenic Emission Inventory System version 3.09 (BEIS3, Vukovich and Pierce, 2002) and, for regions outside Taiwan, by the Model of Emissions of Gases and Aerosols from Nature v2.1 (MEGAN, Guenther et al., 2012). TEDS 10.0 (Taiwan Emission Database System, TWEPA, 2011; https://erdb.epa.gov.tw/,last access: 18 January 2024) was used for domain 3 (d03). To ensure the precision of the multiple dry deposition parameterizations, the present research conducted four simulation scenarios, namely CMAQ_Off_S22, CMAQ_Dust_S22, CMAQ_Dust_E20 and CMAQ_Dust_P22. The CMAQ_Off_S22 scenario did not include the inline dust calculation (Table 3). Meanwhile, the latest refined integrated dust treatment was implemented in the CMAQ_Dust_S22 scenario (Kong et al., 2024). Indeed, both CMAQ_Off_S22 and CMAQ_Dust_S22 used the dry deposition mechanism by Shu et al. (2022). The dry deposition mechanism of Emerson et al. (2020) and Pleim et al. (2022) were implemented in CMAQ_Dust_E20 and CMAQ_Dust_P22 scenarios, respectively.

$V_d$ over the ocean surface has been shown to influence the CTMs in simulating aerosol, particularly $PM_{10}$. The modeled $PM_{10}$ can be increased by reducing $V_d$ by a factor of 10 based on the bare soil measurement (Tav et al., 2018; Ryu et al., 2022). However, the adjusted $V_d$ in estimating the aerosol was too coarse. In Eq. (3), the parameterizations of $R_a$ and $R_b$ determined the magnitude of $V_d$. P22 dry deposition scheme in CMAQv5.4 includes the white-cap effect over the ocean surface, which is related to the particle collection efficiency by impaction, as a function of $R_b$ (Pleim et al., 2022). The impact of the white cap can increase as the wind speed increases, which can be an essential parameterization in simulating transboundary events (Albert et al., 2020). Our findings on dust transport, which was highly related to turbulence, have significant implications for future research and modeling, inspiring further exploration and innovation (Zhang et al., 2022). In CMAQv5.4, $R_b$ is estimated separately by vegetation and non-vegetation type. $R_b$ at the smooth surface (non-vegetation) is related to the surface resistivity of the bare soil and the water layer. Since EAD aerosol particles are mostly uplifted from the bare soil surface layer and the aerosol deposition at the marine boundary layer (Kong et al., 2021), the sensitivity of $R_b$ at the smooth surface impact on the CMAQ dust model simulation can be vital. By assuming $R_b = 1/V_d$, $V_d$ is inversely dependent on $R_b$, we increased $R_b$ to a factor of 10 as P22E01. To further carry out the sensitivity test, we scale $R_b$ by 50 and 100, for P22E02 and P22E03, respectively (Table 3).

**2.4 Ancillary dataset**

PM$_{10}$ (particulate matter ≤10 μm in aerodynamic diameter) and PM$_{2.5}$ (particulate matter ≤2.5 μm in
aerodynamic diameter) concentrations during the dust events in January 2023 were obtained from Lulin
Atmospheric Background Station (LABS; 23.47° N, 120.87° E, 2862 m MSL) and Cape Fuguei (25.30°
N, 121.54° E, 10 m MSL). In addition, the hourly PM$_{10}$ and PM$_{2.5}$ of nearly 100 sites distributed over
mainland China (Fig. S1), covering the period of 12 March-20 April 2021, obtained from the Chinese air
quality online monitoring analysis platform's website (www.aqistudy.cn/). The Modern Era
Retrospective-analysis for Research and Application version 2 (MERRA-2) reanalysis data was used to
demonstrate the spatiotemporal distribution of dust, compare with the air quality model, irrespective of
the influence of clouds. MERRA-2 (Gelaro et al., 2017) is a NASA reanalysis utilizing Goddard Earth
Observing System Data Assimilation System Version 5 (GEOS-5) and covering the data assimilated
system at a native spatial resolution of 0.5 ∘ × 0.625 ∘. Also, Moderate Resolution Imaging
Spectroradiometer (MODIS) Terra satellite images and the level-3 MODIS AOD at 550 nm
(MYD08) were obtained from the U.S. National Aeronautics and Space Administration
(https://worldview.earthdata.nasa.gov/).

## 3 Results and Discussion

### 3.1 Observed air quality and weather conditions

Figure 2 shows the dust outbreak over East Asia, displayed by the MODIS Terra sensor and MODIS
AOD at 550 nm from 22-31 January 2023. The satellite image showed dust induced by a high-pressure
system on 24-25 January (Fig. 2a3, 2a4). The next day, the same region was covered by a thick cloud,
and dust was again widely distributed from 27-30 January 2023. Using MODIS AOD to verify the dust
plume (Han et al., 2012; Kong et al., 2021), the dust plume was distributed in Central China and northern
Taiwan on 24 January 2023. Moreover, the most intense dust plume in the eastern China and East China
Sea region was observed on 27 January. Fig. S2 shows the synoptic weather map across the study domain.
On 22-23 January, the southward high-pressure system was responsible for pushing the pollutant across
the Asian Continent, which is consistent with Chuang et al. (2018) and Kong et al. (2021, 2022, 2024)
(Fig. S2a-b). The high-pressure system that moved southward will then move eastward toward the western
Pacific Ocean (Fig S2c-d). Meanwhile, the high-pressure system on the northwest side again expands in
the southeast direction. The second high-pressure system again pushed the pollutant for the second time
and caused the high pollutant problem on 27 January.
The impact of East Asian dust on the air quality over the high-altitude western Pacific region was
widely discussed (Kong et al., 2022). Two interesting high pollution events at Mt. Lulin (2,862 m above
sea level) during 24-26 Jan and 27-30 January, respectively, are shown in Fig. 3. The latter event was
more intense compared to the earlier one, where the maximum $PM_{10}$ concentration can reach up to 35 μg
$m^{-3}$. Moreover, it was observed that the BC concentrations could reach up to a maximum of 400 ng $m^{-3}$.
Based on the *in-situ* measurement, it was interesting to find the mixing state between dust, BC, and brown
carbon (Fig. 3c). Different from what has been discussed by Kong et al. (2022), the long-range transport
air pollution at the high-altitude not just merely EAD, but also included the anthropogenic pollutant from
mainland China.

## 3.2 Evaluation of CMAQ dust emission and dry deposition parameterizations

Table 4 shows the statistical analysis of $PM_{10}$ and $PM_{2.5}$ concentrations over Cape Fuguei (northern
Taiwan) from 22-31 January under the multiple deposition mechanisms. The threshold of the statistical
index is based on Emery (2001). CMAQ_Off_S22, the $PM_{10}$ simulation presented without the inline dust
calculation, recorded the normalized mean bias (NMB) of -52.81 %. CMAQ_Dust_S22 improved the
simulation over Cape Fuguei (northern Taiwan) by -47.01 % as we included the refined dust treatment
(Kong et al., 2024). However, the improvement is insignificant due to the weak intensity dust episodes
and the limitation due to the excessive deposition mechanism within the model (Kong et al., 2021). Hence,
we expanded the sensitivity simulation to examine the impact of the deposition algorithm on the aerosol
prediction. CMAQ_Dust_E20 simulations utilizing the Emerson et al. (2020) approach increased the
modeled $PM_{10}$ simulation by NMB of -41.9 %.
Instead of $PM_{10}$ simulation, the present study found that the inline dust treatment and deposition
algorithms could influence $PM_{2.5}$ simulation performances. For instance, the modeled $PM_{2.5}$ improved
from -12.63 % (CMAQ_Off_S22) to -8.84 % (CMAQ_Dust_S22). Meanwhile, the deposition algorithm

embedded in CMAQv5.4 has recorded modeled $PM_{2.5}$ by -10.65 % and -15.22 % under CMAQ_Dust_E20 and CMAQ_Dust_P22, respectively. This incident suggested that the East Asian dust from northwest China transported to the Western Pacific Ocean could also carry the anthropogenic emission of East China.

Figure 4 shows the time series of hourly $PM_{10}$ and $PM_{2.5}$ concentrations over Cape Fuguei (northern Taiwan) and LABS (high altitude region) from 22-31 January 2023 under the multiple deposition mechanisms. Generally, all the patterns of $PM_{10}$ simulations were consistent with the observed $PM_{10}$, especially in capturing the peak value. For instance, the maximum observed (CMAQ_Dust_E20) $PM_{10}$ concentrations at the surface during Jan 24 and 27 were 141 (102.6) µg m$^{-3}$ and 114 (163.2) µg m$^{-3}$, respectively. A similar time-series pattern was found for the $PM_{2.5}$ simulation (Fig. 4b).

More importantly, the CMAQ model performance over the high-altitude region needed to be carried out and discussed. The biomass-burning episode of the northern PSEA over Mt. Lulin has been finely correlated by plume rise injection (Chuang et al., 2016; Ooi et al., 2021). From Fig. 4c, the modeled $PM_{10}$ pattern for CMAQ_Off_S22 could not correlate well with observed $PM_{10}$ over Mt. Lulin, with a poor correlation of 0.30. The correlation was increased for CMAQ_Dust_S22 (0.54), CMAQ_Dust_P22 (0.46), and primarily well performed for CMAQ_Dust_E20 (0.55). The modeled result was somehow consistent with the surface $PM_{10}$ simulation at Cape Fuguei. The high observed $PM_{10}$ episodes during 27-28 January with a maximum of 34.5 µg m$^{-3}$ was only 53.3 % higher than CMAQ_Dust_E20 of 22.5 µg m$^{-3}$. For the CMAQ $PM_{2.5}$, the simulation generally underestimated the observed $PM_{2.5}$. It's worth noting that E20, in particular, showed exceptional performance in the $PM_{10}$ simulation compared to other dry deposition schemes under the refined dust scheme. This underscores the potential effectiveness of E20 in managing $PM_{10}$ particulate matter. However, the $PM_{2.5}$ simulations showed only marginal changes, regardless of whether it was a surface or high-altitude simulation.

During the spring of 2021, a series of dust storms (15 March, 27 March, and 18 April) occurred over the Gobi area, with one of the most significant dust storms in the past decade (15 March, the "3.15" dust storm hereafter) causing environmental impact over the continental (Jin et al., 2022; Gui et al., 2022;

He et al., 2022; Liang et al., 2022; Tang et al., 2022). More interestingly, one of the multiple dust storm
episodes reached western Pacific Ocean due to the extreme typhoon episode (Kong et al., 2024). Hence,
we intend to re-emphasize the precision of various deposition schemes on the CMAQ for the recent dust
storm episode over the Asian Continental highlighted by Kong et al. (2024). We evaluated the CMAQ
simulations with the different dry deposition schemes for the 40-day sensitivity test on 12 March-20 April
2021 against measured $PM_{10}$ and $PM_{2.5}$ concentrations across the observation sites in mainland China
(Table 5). The observation sites used for the model comparison are marked in Fig. S1. Generally, the
evaluation results for Taiwan and mainland China were consistent. During the 40 days of Spring 2021,
the CMAQ $PM_{10}$ of NMB was the highest for Off_S22 (NMB = -75.00 %), followed by Dust_S22 (-45.97
%). The latest inline dust emission scheme embedded with E20 dry deposition scheme for $PM_{10}$ was well
performed by NMB of -25.43 %, compared to the Dust_P22 (-59.82 %). For the $PM_{2.5}$ simulation,
Dust_S22 has been improved from Off_S22, and Dust_S22 was slightly better than Dust_E20 and
Dust_P22.

302        Figure 5 shows the scatter plot of simulated and observed PM across mainland China. The

correlation coefficient (R), a factor of two (FAC2), and the mean observed and simulated PM are marked
in Figure 5. The modeled $PM_{10}$ without the dust scheme had the lowest correlation. Among all of these
simulations, Dust_E20 performed the best correlation (R > 0.3) compared to Dust_S22 and Dust_P22.
However, for $PM_{2.5}$, the correlation between the model and measured values was similar for all the dry
deposition schemes. The statistical index of FAC2 was used in the present work since either low or high
outliers less influence it (Chan and Hanna, 2004). The dataset is reliable for FAC2 values between 0.5
and 2.0, with the ideal model of 1.0. The simulated $PM_{10}$ by Dust_E20 performed well, with a nearly
perfect value of 1.1. Meanwhile, the $PM_{2.5}$ by Dust_S22 simulation was slightly better than Dust_E20 but
much better than the other experiments.

312        The comparison of AOD between CMAQ and MODIS for the three dust storm episodes: 14-16

March 2021 ("3.15" dust storm), 26-28 March 2021 ("3.27" dust storm), and 17-19 April 2021 was shown
("4.18" dust storm) (Table 5). Overall, CMAQ Dust_E20 above 30°N has evaluated well the MODIS
AOD by NMB of -26.2 %, as compared to S22 (-32.0 %) and P22 (-35.8 %). The CMAQ AOD by E20
during the most intense Super Dust Storm in 3.15 has significantly improved over northern China, the
dust source region, as shown in the red dash rectangular box (Fig. S3). Additionally, the modeled AOD
by E20 over the western Pacific Ocean (shown in red dash rectangular box) increased in episode 4.18,
reporting a value of 0.7 compared to 0.5 by S22. Significantly, the E20 deposition scheme has primarily
enhanced the $PM_{10}$ prediction over the marine boundary layer, addressing the model uncertainty due to
the typhoon mentioned by Kong et al. (2024) and demonstrating the practical implications of our research.

322       The present work is consistent with the dust scheme in the WRF-Chem, where the dust loading is

very sensitive to the dry deposition schemes and dust emission schemes, especially over the downwind
region (Zeng et al., 2020). Fig. 6 shows the CMAQ estimated averaged mean $PM_{10}$ and $PM_{2.5}$ in January
2023 and Spring 2021 for the Off_S22 and its corresponding change by Dust_S22, Dust_E20, and
Dust_P22, respectively. Generally, the spatial distribution of the high $PM_{10}$ concentrations by > 50 µg m$^{-3}$
$^{3}$ was distributed over northwest China, which is the dust source region's location, consistent with the
simulation suggested by Kong et al. (2021, 2022, 2024). Such high particulate matter dissipated to east
China, indicating the transport pathway in the southeastern direction towards the western Pacific (Fig. 6a,
h). The larger $PM_{10}$ distribution by E20 than S22 and P22 over northwest China, meaning E20
successfully increased the $PM_{10}$ concentrations. Another fascinating fact about E20 was that the $PM_{10}$
increased over the southern South China Sea (Fig. 6b). For the modeled $PM_{2.5}$ concentrations, the high
concentration was distributed over the Asian Continental under all dry deposition mechanism. Similar to
the trend of $PM_{2.5}$ simulations in Taiwan (as shown in Fig. 4), the spatial distribution of the modeled
$PM_{2.5}$ was identical to that of all dry deposition schemes. The result implies the significant impact of dry
deposition on the EAD simulation's dust model, displaying the positive relationship between dust
deposition and $PM_{10}$ concentrations (Zhang et al., 2017).
**3.3 Impact on the CMAQ ambient particle concentrations**
Figure 7a shows the boxplot of the averaged simulated $V_d$ for the Aitken, accumulation, and coarse
particles modes under multiple deposition schemes in January 2023 (S22_2023, E20_2023, and
P22_2023) and in Spring 2021 (S22_2021, E20_2021, and P22_2021). These different dry deposition

treatments have a substantial impact on the aerosol profile, altering the ambient total dry deposition regionally. For instance, the median deposition velocity of S22_2023, E20_2023, and P22_2023 of the Aitken (accumulation) modes particle were 0.069 (0.020) cm s$^{-1}$, 0.039 (0.014) cm s$^{-1}$ and 0.034 (0.029) cm s$^{-1}$, respectively. The E20 simulation median $V_d$ decreased by -12.65 % for coarse-mode particles compared to S22. Also, the 75$^{th}$ percentile $V_d$ of the coarse mode has been significantly reduced by -32.13 %. On the other hand, P22 showed a different simulation by the median $V_d$ increment of 71.38 %. These findings suggest that the choice of dry deposition treatment can significantly influence the distribution and concentration of aerosols in the atmosphere, with potential implications for air quality and climate.

As shown in Figure 7a, the results during the spring of 2021 are similar to those for January 2023 in comparing the dry deposition schemes. Notably, the $V_d$ of the coarse mode for E20_2023 and E20_2021 was lowest compared to the other dry deposition schemes. Contrary, the accumulation and coarse mode by P22 were the highest. The result was consistent with the best simulated PM$_{10}$ by E20 in 2023 and 2021 displayed in Table 4 and 5, respectively. The lowest $V_d$ of the coarse mode particle was responsible for reducing the PM$_{10}$ simulation underestimation, consistent with the simulation by Ryu and Min (2022). The slow $V_d$ means the total loss of aerosol to the surface has been minimized, leading to increased aerosol concentration. In addition, the spatial distribution of dust emissions could significantly influence the aerosol deposition velocity. The total dust emission in Spring 2021 was of a much higher magnitude and wider spatial distribution than in January 2023 (Fig. 7b, c). This led to a slow $V_d$ in the coarse mode, particularly, causing more dust loading during the multiple dust storms in Spring 2021 than the regular dust episode recorded in January 2023. This finding is consistent with Zeng et al. (2020), which emphasized the sensitivity of different dust emissions on dry deposition schemes. However, it's important to note that the research was only conducted in one particular short period. On the other hand, this work has highlighted the distinct dust emission according to EAD intensity impacting the various dry deposition schemes. These implications are crucial for understanding the behaviour of aerosols in the atmosphere and their significant impact on air quality.

We estimated the CMAQ averaged particle modes for the S22_2023, E20_2023, and P22_2023 dry deposition scheme (Fig. 8). For S22_2023, we found that high $V_d$ corresponding to the Aitken and

accumulation modes distributed mainly over most of the CMAQ domain, which was most evident over Asian continent (Fig 8a, 8d). Meanwhile, the magnitude of $V_d$ distribution was the most significant over the western Pacific Ocean by S22_2023 and the least for E20_2023 (dash rectangular box in Fig. 8d, e, f). For the coarse mode particles, the $V_d$ was the lowest for E20_2023 compared to S22_2023 and P22_2023, particularly over the ocean area near northeast China, Japan, and Korea (white-dash rectangular box in Fig. 8d, e, f). This leads to a significant deposition over the downwind region, causing less $PM_{10}$ simulated by P22_2023 and S22_2023 than E20_2023. A previous study proposed the $V_d$ for the aerosol at the water surface was associated with the CTM uncertainly at the downwind region (Kong et al., 2021, 2024; Ryu and Min, 2022). The $V_d$ at land surface was generally higher than at water surfaces. Interestingly, the coarse mode $V_d$ at the water surface for E20_2023 (0.060 cm s$^{-1}$) was lower than S22_2023 (0.085 cm s$^{-1}$) and P22_2023 (0.116 cm s$^{-1}$), respectively, suggesting that E20_2023 deposition schemes could minimize the excessive deposition over the marine boundary layer (Table 6). Such minimal deposition velocity distributing over a large part of the western Pacific Ocean, including the Sea of Japan, Yellow Sea, East China Sea, and South China Sea, might be responsible for reducing the modeled $PM_{10}$ underestimation over Taiwan (Fig. 8h), as mentioned by Kong et al. (2021).

To better understand the behavior of the $V_d$ during the 40-day simulation of Spring 2021 corresponding to the aerosol simulation, we visualized the CMAQ averaged particle modes for the S22_2023, E20_2023, and P22_2023 dry deposition scheme (Fig. 9). The $V_d$ of the coarse mode particles for E20_2021 was the lowest among the others over the ocean area, which shows similarity as E20_2023 (Fig. 9g, h, i). As mentioned by Kong et al. (2024), one of the continuous EAD episodes was related to the typhoon. The strong wind speed and extreme precipitation due to the intense anticyclonic system caused nearly zero dust simulation. In S22_2021 (0.060 cm s$^{-1}$) and P22_2021 (0.070 cm s$^{-1}$), the model suggested high coarse mode $V_d$ at the western Pacific Ocean. In E20_2021, the $V_d$ (0.053 cm s$^{-1}$) is lower than the rest of the dry deposition mechanism, particularly the area affected by typhoon (black-dash rectangular box). This means that the E20 dry deposition has reduced the uncertainty of the excessive dust loss at the marine boundary layer. Figure 6 (g, h, i) shows more simulated mineral dust at the western Pacific by E20 than S22 and P22 during the spring of 2021.

### 3.4 CMAQ of dust and black carbon synoptic pattern at the upper level

Black carbon (BC), often known as elemental carbon, released from the biofuels, fossil fuels and biomass burning, has been proven to impact the radiative budget and regional climate (Ramanthan, V and Carmicheal, 2008; Pani et al., 2016, 2020). In the meantime, China has been a significant contributor to global anthropogenic BC emission, particularly in the cities of the northern part (Xiao et al., 2023; Wang et al., 2024). During the severe dust episodes in the spring of 2023, the contribution of black carbon brought by EAD was captured in North China (Wang et al., 2024). As depicted in Fig. 2, the transboundary episode observed in the upper level of Taiwan during this event could be the mixing of the natural dust and anthropogenic haze episodes, which demonstrates the consistency. Additionally, blending mineral dust with anthropogenic transport due to the north easterly wind, a wind that blows from the northeast, has been a subject of extensive discussion (Lin et al., 2007, 2012; Li et al., 2012). During the EAD, the dust from the Gobi Desert that was transported towards the western Pacific region could also carry anthropogenic aerosol, contributing to different levels of pollutant concentration. However, the distinct transport pathway at the high altitude between both aerosol types is a topic that has received less attention but is of significant importance to our understanding of atmospheric dynamics.

Figure 10 illustrates mineral dust and BC concentration's spatial and temporal distribution under the CMAQ_Dust_E20 scenario at 700 hPa from 24-31 January. The model reveals a high proportion of modeled dust aerosol (red dash circle) at the source region, indicating an uplift from the surface to 700 hPa (Fig. 10a). This uplift, driven by the strong pressure gradient at the surface and the 'eastward moving trough system' at the upper level (700 hPa), is a key factor in the eastward and southward transfer of the dust (Fig. 10b). The high dust fraction reappears at the source region (Fig. 10c) and is transported eastwardly by the similar upper-level trough (Fig. 10d), causing a long dust belt at 15°N, distributing over central Asia continental, Taiwan Straits, Taiwan and large part of western Pacific Ocean. (Fig. 10e). On 29 January, the model of E20 clearly predicted that the dust plume moved in the southward direction toward the South China Sea (Fig. 10f). The dust aerosol was left distributed at a certain part of the northern South China Sea and the Philippine Sea until it totally dissipated (Fig. 10g, h). This interesting result

suggests the possible EAD at the longer distance at the upper level, which is a topic for further
investigation.
The southward high-pressure system responsible for the long-range transport haze episode has
been widely discussed (Chuang et al., 2008; Kong et al., 2021)—however, the upper-level transboundary
transport needs to be addressed more. While focusing on CMAQ_Dust_E20, we attempted to characterize
the long-range transport of modeled black carbon at the upper level (700 hPa) (Fig. 10i-p). As shown in
Fig. 10(i), the modeled black carbon concentration is shown to be significantly distributed at central
China. The black carbon transport pattern followed the eastward-moving trough system as the plume
moved eastward and southward (Fig. 10m, n). Interestingly, the long black carbon belt is consistent with
the long dust belt, as shown in Fig. 10(e, f). For instance, both modeled dust and BC were distributed at
the western Pacific Ocean (Fig. 10e, f, m, n) and South China Sea (Fig. 10g, o). This means that the BC
due to the anthropogenic emission and the natural EAD shared a similar transport pattern at the upper
level, driven by the trough system. Such consistency has been verified by the MERRA-2 dust and BC
mass column over the region (red dash rectangular in Fig. S4).
The dust aerosol vertical profiles (Fig. 11) show a significant distribution of the large dust fraction
over the Asian Continent under all simulation scenarios (Fig. 11a1-e1), as indicated by the transect drawn
in Fig. 1. The westerly winds, depicted in Fig. 10, facilitated the eastward transport of the aerosol plume
towards the western Pacific Ocean, where it accumulated along the 700 hPa altitude. Another plume was
observed across the ocean on the east side of Taiwan Island (Fig. 11b1). On 27 January, showed another
substantial fraction of dust covering the Asian Continent and Western Pacific Ocean, with significantly
higher dust concentrations compared to Fig. 11a1. The plume distributed eastward exhibited a clear dust
dome (Fig. 11a5-e5). These findings have important implications for understanding and predicting dust
aerosol transport patterns and their potential environmental impact.
The vertical profile of the modeled BC mirrors the transport pattern of mineral dust, as shown in
Fig. 12. A transparent BC dome was distributed along 700 hPa, echoing the pattern observed for dust.
This simulation suggests the consistency of the "double dome" mechanism of Asian dust and biomass

burning episodes (Dong et al., 2018; Huang et al., 2019). The potential warming effect of such a mechanism is a topic ripe for future studies. However, it's important to note that the dust dome contains a higher fraction of concentrations than the black carbon dome. The present simulation suggests that dust aerosol can reach up to 500 hPa, which is consistent with Kong et al. (2021). On the other hand, the black carbon plume was slightly lower, with approximately 600 hPa of the maximum height under the same meteorological condition. This section, which discusses the similarity and distinctiveness of natural dust and anthropogenic aerosol at the upper level, highlights the need for further study. The present simulation did not consider the two-way coupling model, and it is strongly suggested for future research.

Table 7 shows the modeled deposition and mass concentration for different simulation scenarios in January 2023. The simulation of the wet deposition and mass concentration for dust aerosol was the highest by E20. This is consistent with the globally averaged aerosol number concentrations over the ocean for the large size particle (Emerson et al., 2020). Contrary, P22 was the lowest in simulated wet deposition and mass concentration. P22 could increase the accumulation mode's $V_d$ and reduce the $PM_{2.5}$ over CONUS, which is similar to the present result (Pleim et al., 2022). Moreover, the present simulation by P22 showed the highest $V_d$ of the coarse mode that leads to the less simulated $PM_{10}$. P22 revised the impaction collective efficiency, which is the parameterization of $R_b$. In order to understand the sensitivity of $R_b$ on CMAQ simulation, the $R_b$ has been scaled up, as shown in Table 3. Generally, the increment of $R_b$ has gradually increased the wet deposition (surface mass concentration) by 13.6 (45.8) %, 25.2 (83.3) %, and 28.2 (93.7) %, under P22E01, P22E02 and P22E03, respectively. In addition, the increment intensity at the surface was higher than at the upper level. The simulated dust at western Pacific Ocean responding to the different dry deposition schemes was shown during 27 January in Figure 11 (red-dash rectangular box). As $R_b$ increased by P22E01 and P22E03, the simulated $PM_{10}$ by base scheme P22 (~30 $\mu g\ m^{-3}$), has increased to ~40 $\mu g\ m^{-3}$ and ~50 $\mu g\ m^{-3}$, respectively. It is worth noted that P22E03 simulated a similar dust concentration as E20, indicating the importance of revising the $R_b$. On the contrary, the wet deposition and mass concentration were most significant for modeled BC under the S22 dry deposition scheme (Table 7). P22E01 only showed a minor increment, but it was nearly identical for P22E02 and P22E03 compared to P22.

## 4.0 Summary and Conclusions

The chemical transport model is considered sensitive to the dry deposition parameterization besides the dust emission treatment. The present study demonstrates the impact of the dry deposition parameterizations (S22, E20, and P22) on aerosol performance in East Asia. It provides a significant analysis of the transboundary transport of East Asian Dust to Taiwan from a 22-31 January 2023 case study and multiple heavy dust storm episodes from 12 Mar-20 Apr 2021. Incorporating the latest dust emission treatment (Kong et al., 2024) into the CMAQ slightly improved the model performance to -47.01 % from -52.81 %. By implementing the E20 dry deposition scheme, characterized by calibrating the collection efficiency by Brownian diffusion and interception, the CMAQ simulation of the surface $PM_{10}$ has been improved by NMB of -41.9 %, as compared to the dry deposition proposed by P22 (-53.90 %). Moreover, the modeled $PM_{10}$ pattern by E20 at the upper level (700 hPa) was mainly consistent with the observed $PM_{10}$, especially in capturing the peak value. The dry deposition of E20 was correlated well with the high altitude in situ by 0.55, as compared to S22 (0.54) and P22 (0.46). On the contrary, simulated surface $PM_{2.5}$ by S22 has been improved to -8.84 % from -12.63 % after using the latest dust treatment, and slightly better performance than E20 (-10.65 %) and P22 (-15.22 %). Additionally, the simulations of the multiple dust episodes in spring 2021 were re-constructed to evaluate the CMAQ performance over the Asian Continental. The E20 dry deposition scheme outperformed the others with the lowest NMB value in simulating $PM_{10}$ (-25.4 %) and AOD (-26.2%). For the modeled $PM_{2.5}$, S22 performed slightly better than E20, with NMB of -36.29 % and -37.5 %, respectively.

The previous CMAQ model, modulated by Kong et al. (2021; 2024), showed excessive deposition at the marine boundary layer, leading to underestimating the modeled surface $PM_{10}$. However, using the E20 scheme over the entire model domain, our updated model has the lowest $V_d$. This precise reduction of $V_d$ of the coarse mode particle, responsible for reducing the $PM_{10}$ simulation underestimation, has not just minimized, but effectively minimized the total loss of aerosol to the surface, leading to a concentration increment. Furthermore, the low-lying modeled $V_d$ across the water surface by E20 could be crucial in reducing the excessive aerosol deposition over the ocean layer.

It is worth revealing that the transboundary transport of EAD from the Asian continent towards the western Pacific Ocean at the upper level was associated with the eastward moving trough system. Such transport mechanisms have been found to bring along black carbon aerosol, which is primarily the main element of China's human-made emissions. More interestingly, both aerosol profiles created a "long dust-black carbon belt" along the 15°N. The 'double dome mechanism', a concept proposed by Huang et al. (2019) that depicts the superposition of the two aerosol types, was also simulated in the present study. Besides the similarity of both, the discrepancy in the case of the aerosol deposition and mass concentration was shown. By comparing the base P22 scheme to the revised scheme (P22E01-P22E03), the dust aerosol increased significantly and marginally by the black carbon. This study highlights the importance of dry deposition schemes for the modeled dust and black carbon concentration and provides a reference for better dry deposition schemes in CTMs over East Asia.

We noted that the improved model simulation for EAD relied on dust emission, dust deposition, and transport processes. The dust emission treatment was proven sensitive to the CMAQ model performance in East Asia (Dong et al., 2018; Liu et al., 2021; Kong et al., 2024). In addition, the CTM performance can be attributed to the dust emission schemes and the dry deposition schemes (Zeng et al., 2020). In other words, different dust emission schemes may impact the $V_d$ and dust loading, which reacts differently to model performance. The present research, which is a complex examination, is of significant importance as it primarily focuses on which dry deposition scheme can improve the most recent updated dust emission model. Therefore, the sensitivity of the dust emission parameterizations or approaches, including surface roughness, land surface, soil texture, and types on the dry deposition scheme, underscores the need for a comprehensive understanding and is proposed for future studies.

Finally, it is necessary to point out that the dry deposition on the EAD is closely associated with the $PM_{10}$ concentration (Zhang et al., 2017). Nevertheless, it has been shown that there are other atmospheric processes related to the air quality over the Western Pacific, including transboundary haze, biomass burning, and local emission (Chuang et al., 2020; Ooi et al., 2021; Chang et al., 2023). These complex phenomena could cause variations of $PM_{2.5}$, ozone, and the corresponding primary pollutant. Hence, the

role and response of the dry deposition scheme in the CMAQ should be paid attention to in the future for compressive understanding and model improvement. This research enhances our understanding of dust emission and dry deposition models and provides valuable insights for improving air quality models, which is crucial for environmental and public health management.

**Data Availability**

MERRA-2 data are available online through the NASA Goddard Earth Sciences Data Information Services Center (GES DISC; https://disc.gsfc.nasa.gov; last access: 01 August 2024). MODIS data used in this study are available at https://asdc.larc.nasa.gov/(last access: 01 August 2024). The observational data at LABS can be ordered by contacting corresponding authors.

**Author Contribution**

**Steven Soon-Kai Kong**: Conceptualization; Data curation; Formal analysis; Investigation; Methodology; Software; Validation; Visualization; Writing – original draft; Writing – review and editing.

**Joshua S. Fu**: Conceptualization; Investigation; Methodology; Formal analysis; Writing – review and editing.

**Neng-Huei Lin**: Conceptualization; Visualization; Supervision; Funding acquisition; Resources; Writing – review and editing.

**Guey-Rong Sheu**: Funding acquisition; Resources.

**Wei-Syun Huang:** Data curation; Software.

**Competing Interests**

Some authors are members of the editorial board of journal ACP.

**Acknowledgments**

We acknowledged the National Science and Technology Council of Taiwan, under Project No. NSTC113-2811-M-008-045 for supporting the research. We also acknowledged the staff at LABS, and EPA Taiwan for the provision of the ground-based measurement datasets. We are also thankful to MERRA-2 and MODIS for the satellite product.

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

**Table 1.** Detailed mechanism expression relating the three dry deposition schemes.

| Schemes | Surfaces | S22 (CMAQv5.3 and beyond) | E20 | P22 |
|---|---|---|---|---|
| $V_d$ | | $f_{veg}V_{d\ vegetated} + (1-f_{veg})V_{d\ smooth}$ | $f_{veg}V_{d\ vegetated} + (1-f_{veg})V_{d\ smooth}$ | $f_{veg}V_{d\ vegetated} + (1-f_{veg})V_{d\ smooth}$ |
| $R_b$ | Vegetated | $\dfrac{1}{f_{veg}((\max{(LAI,1.0)})u_*(E_B+E_{Im}))}$ | $\dfrac{1}{wet*E_{Tot\ veg}+(1-wet)*E_{Tot\ veg}*R1}$ | $\dfrac{1}{f_{veg}((\max{(LAI,1.0)})u_*(E_B+E_{Im}))}$ |
| $R_b$ | Smooth | $\dfrac{1}{u_*(E_B+E_{Im})}$ | $\dfrac{1}{wet*E_{Tot\ smth}+(1-wet)*E_{Tot\ smth}*R1}$ | $\dfrac{1}{BAI.u_*(E_B+E_{Im})}$ |
| $E_B$ | Vegetated | $Sc^{-2/3}$ | $C_B Sc^{-2/3}$ | $C_{IB}Sc^{-2/3}$ |
| $E_B$ | Smooth | $Sc^{-2/3}$ | $C_B Sc^{-2/3}$ | $f_{wc}\dfrac{u_*}{U_{10}}+(1-f_{wc})C_{IB}Sc^{-2/3}$ |
| $E_{Im}$ | Vegetated | $\dfrac{St^2}{St^2+1}$ | $C_{Im}(\dfrac{St}{St+\alpha})^{1.7}$ | $f_{micro}\dfrac{Sth^2}{Sth^2+1}+(1-f_{micro})\dfrac{St1^2}{St1^2+1}$ |
| $E_{Im}$ | Smooth | $\dfrac{St^2}{St^2+400}$ | $C_{Im}(\dfrac{St}{St+100})^{1.7}$ | $10^{-3/St}$ |
| $E_{In}$ | Vegetated | 0 | $C_{In}\left(\dfrac{d_p}{A}\right)^{0.8}$ | 0 |
| $E_{In}$ | Smooth | 0 | 0 | 0 |

$V_{d\ vegetated}$ = deposition velocity over the vegetative surface: $V_{d\ vegetated} = \dfrac{V_g}{1-\exp{(-V_g(R_a+R_{b\ vegetated}))}}$
$V_{d\ smooth}$ = deposition velocity over the smooth surface: $V_{d\ smooth} = \dfrac{V_g}{1-\exp{(-V_g(R_a+R_{b\ smooth}))}}$
$f_{veg}$ = grid scale vegetation-coverage fraction
$E_B$ = Brownian diffusion efficiency
$E_{Im}$ = Impaction efficiency
$E_{In}$ = Interception efficiency
Sc = Schmidt number
$St$ = Stokes number
wet = Wet surface
$E_{Tot\ veg}$ = veg_ustar*$(E_B + E_{Im} + E_{In})$
$E_{Tot\ smth}$ = 3.0*ustg*$(E_B + E_{Im})$
R1 = Bounce correction term by Slinn (1982).
$C_B$ = Brownian collective coefficient: 0.2
$C_{Im}$ = Impaction collective coefficient: 0.4
$C_{In}$ = Interception collective coefficient: 2.5
α = Empirical constant
LAI = Leaf area index
BAI = Building area index
$C_{IB}$ = 1.0/3.0
$f_{wc}$ = Whitecap surface fraction
$f_{micro}$ = Total impaction fraction from the microscale features
$u_*$ and $U_{10}$ = Frictional velocity and wind speed at 10 m (ms$^{-1}$)
$St1$ and $Sth$ = Obstacle characteristic dimensions for the leaf hairs and microscale roughness on leaves
**Table 2.** Model settings.

| Model setting | Descriptions |
| --- | --- |
| Period | 12 March-20 April 2021 and 22-31 January 2023 |
| Domain | d01, d02, and d03 with 45 KM, 15 KM, and 5 KM of the resolutions, respectively |
| Boundary condition | NCEP FNL lateral boundary condition |
| Surface and land surface model | NOAH |
| Numerical weather model | WRF v40, including grid and observation nudging at d01. |
| Chemical transport model | CMAQ v5.4 |
| Gas-phase chemistry and aerosol mechanism | CB06e51 + AE7 |
| Emission Inventory | MICS-ASIA III emission in 2023, adjusted from the emission in 2017 (Zhang et al., 2018) based on the OMI-NO$_x$ satellite (Huang et al., 2021). |
| Online dust treatment | The refined windblown dust treatment suggested by Kong et al. (2024). |
| Dry deposition option | STAGE (S22, E20 and P22). |


**Table 3.** Simulation scenarios used in this present study.

| Experiments | Online dust emission treatment by Kong et al. (2024) | Dry deposition treatment | Surface resistance ($R_b$) at the smooth surface |
|---|---|---|---|
| CMAQ_Off_S22 | Off | S22 | Default |
| CMAQ_Dust_S22 | On | S22 | Default |
| CMAQ_Dust_E20 | On | E20 | Default |
| CMAQ_Dust_P22 | On | P22 | Default |
| CMAQ_Dust_P22E01 | On | P22 | Increased by a factor of 10 |
| CMAQ_Dust_P22E02 | On | P22 | Increased by a factor of 50 |
| CMAQ_Dust_P22E03 | On | P22 | Increased by a factor of 100 |


**Table 4.** Statistical evaluation for $PM_{10}$ and $PM_{2.5}$ concentrations during 22-31 January 2023 for Cape
Fuguei under the multiple simulation scenarios.

| | Benchmark | CMAQ | | | | | | |
|---|---|---|---|---|---|---|---|---|
| | | Off_S22 | Dust_S22 | Dust_E20 | Dust_P22 | Dust_P22E01 | Dust_P22E02 | Dust_P22E03 |
| | | **$PM_{10}$** | | | | | | |
| MeanObs | | 49.97 | 49.97 | 49.97 | 49.97 | 49.97 | 49.97 | 49.97 |
| MeanMod | | 23.58 | 26.48 | 29.04 | 23.04 | 25.99 | 27.36 | 27.69 |
| NMSE | | 0.66 | 0.56 | 0.49 | 0.71 | 0.57 | 0.53 | 0.52 |
| NMB | ± 85% | -52.81 | -47.01 | -41.90 | -53.90 | -47.99 | -45.24 | -44.58 |
| R | > 0.35 | 0.43 | 0.46 | 0.52 | 0.42 | 0.48 | 0.51 | 0.52 |
| NMBF | | -1.12 | -0.89 | -0.72 | -1.17 | -0.92 | -0.83 | -0.80 |
| | | **$PM_{2.5}$** | | | | | | |
| MeanObs | | 15.52 | 15.52 | 15.52 | 15.52 | 15.52 | 15.52 | 15.52 |
| MeanMod | | 13.56 | 14.15 | 13.86 | 13.16 | 13.26 | 13.22 | 13.20 |
| NMSE | | 0.30 | 0.30 | 0.29 | 0.31 | 0.30 | 0.30 | 0.30 |
| NMB | ± 85% | -12.63 | -8.84 | -10.65 | -15.22 | -14.54 | -14.80 | -14.92 |
| R | > 0.35 | 0.50 | 0.53 | 0.53 | 0.52 | 0.53 | 0.53 | 0.53 |
| NMBF | | -0.14 | -0.20 | -0.12 | -0.18 | -0.17 | -0.17 | -0.18 |

Note: the definition of the statistical formulas NMSE: Normalized Mean Square Error; NMB: Normalized
Mean Bias; R: Correlation Coefficient and NMBF: Normalized Mean Bias Factor









**Table 5.** CMAQ evaluation for $PM_{10}$ and $PM_{2.5}$ against the averaged 100 observation sites across mainland China (Fig. S1) and AOD against MODIS daily observation near the dust source region (above 30°N) with Normalized Mean Bias (NMB) under the multiple simulation scenarios (Fig. S3). Spring 2021, 3.15, 3.27, and 4.18 represent the evaluation period by 12 March-20 April 2021, 14-16 March 2021, 26-28 March 2021, and 17-19 April 2021, respectively.

| Parameters | Period | CMAQ | | | | | | |
|---|---|---|---|---|---|---|---|---|
| | | Off_S22 | Dust_S22 | Dust_E20 | Dust_P22 | Dust_P22E01 | Dust_P22E02 | Dust_P22E03 |
| $PM_{10}$ | Spring 2021 | -75.00 | -45.97 | -25.43 | -59.82 | -45.09 | -35.42 | -32.92 |
| $PM_{2.5}$ | Spring 2021 | -55.56 | -36.29 | -37.50 | -42.47 | -41.20 | -41.51 | -41.66 |
| | | | | | | | | |
| AOD | 3.15 | -80.49 | -46.41 | -38.97 | -48.45 | -44.80 | -41.66 | -40.80 |
| | 3.27 | -80.92 | -41.84 | -36.39 | -44.52 | -41.60 | -39.30 | -38.72 |
| | 4.18 | -83.09 | -7.83 | -3.20 | -14.52 | -9.45 | -7.18 | -6.67 |
| | Mean AOD | -81.50 | -32.03 | -26.19 | -35.83 | -31.95 | -29.38 | -28.73 |

**Table 6.** Average deposition velocity in January 2023 (S22_2023, E20_2023, and P22_2023) and Spring 2021 (S22_2021, E20_2021, and P22_2021) for Aitken, Accumulation, and Coarse modes over land and ocean boundary layer, respectively.

| Dry deposition schemes (cm s$^{-1}$) | Aitken | | Accumulation | | Coarse | |
|---|---|---|---|---|---|---|
| | Land | Ocean | Land | Ocean | Land | Ocean |
| S22_2023 | 0.219 | 0.117 | 0.120 | 0.064 | 0.078 | 0.085 |
| E20_2023 | 0.090 | 0.074 | 0.065 | 0.040 | 0.139 | 0.060 |
| P22_2023 | 0.085 | 0.062 | 0.072 | 0.043 | 0.290 | 0.116 |
| | | | | | | |
| S22_2021 | 0.308 | 0.100 | 0.109 | 0.042 | 0.077 | 0.060 |
| E20_2021 | 0.139 | 0.063 | 0.063 | 0.026 | 0.142 | 0.053 |
| P22_2021 | 0.119 | 0.047 | 0.072 | 0.025 | 0.265 | 0.070 |

**Table 7.** Model averaged dry, wet deposition and mass concentration for dust and BC aerosols in January 2023 (10-days averaged) for different simulation scenarios.

| Dust (ASOIL) | Dry deposition (mg m$^{-2}$) | Wet deposition (mg m$^{-2}$) | Mass concentration at the surface (µg m$^{-3}$) | Mass concentration at 700 hPa (µg m$^{-3}$) |
|---|---|---|---|---|
| S22 | 0.267 | 0.112 | 6.34 | 3.62 |
| E20 | 0.167 | 0.136 | 10.25 | 4.40 |
| P22 | 0.300 | 0.103 | 4.79 | 3.56 |
| P22E01 | 0.243 | 0.117 | 7.00 | 3.79 |
| P22E02 | 0.196 | 0.129 | 8.78 | 4.13 |
| P22E03 | 0.183 | 0.132 | 9.28 | 4.22 |
| BC (AECI + AECJ) | Dry deposition (µg m$^{-2}$) | Wet deposition (µg m$^{-2}$) | Mass concentration at the surface (ng m$^{-3}$) | Mass concentration at 700 hPa (ng m$^{-3}$) |
| S22 | 5.13 | 50.49 | 492 | 60.04 |
| E20 | 8.09 | 48.27 | 471 | 57.73 |
| P22 | 17.79 | 40.96 | 411 | 50.95 |
| P22E01 | 16.88 | 41.64 | 415 | 51.23 |
| P22E02 | 16.82 | 41.67 | 415 | 51.27 |
| P22E03 | 16.82 | 41.67 | 415 | 51.27 |

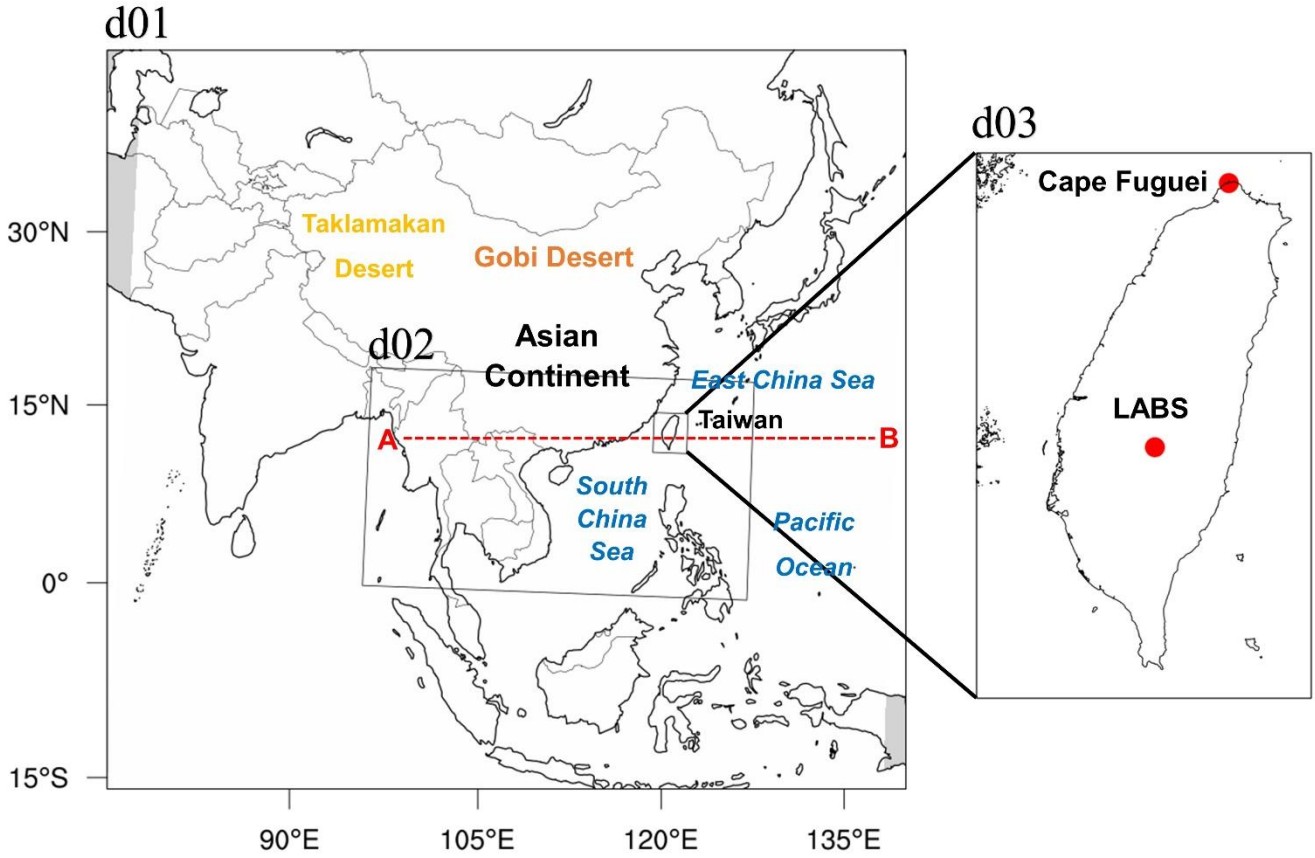


**Figure 1:** Modeling domain configuration in East Asia. Ground-based air quality stations in Taiwan at
Cape Fuguei and Lulin Atmospheric Background Station (LABS) are shown in the zoomed panel. The
red dash line (A➤B) represents the transects that the aerosol plumes traveled along in this study and that
are discussed in Section 3.4;

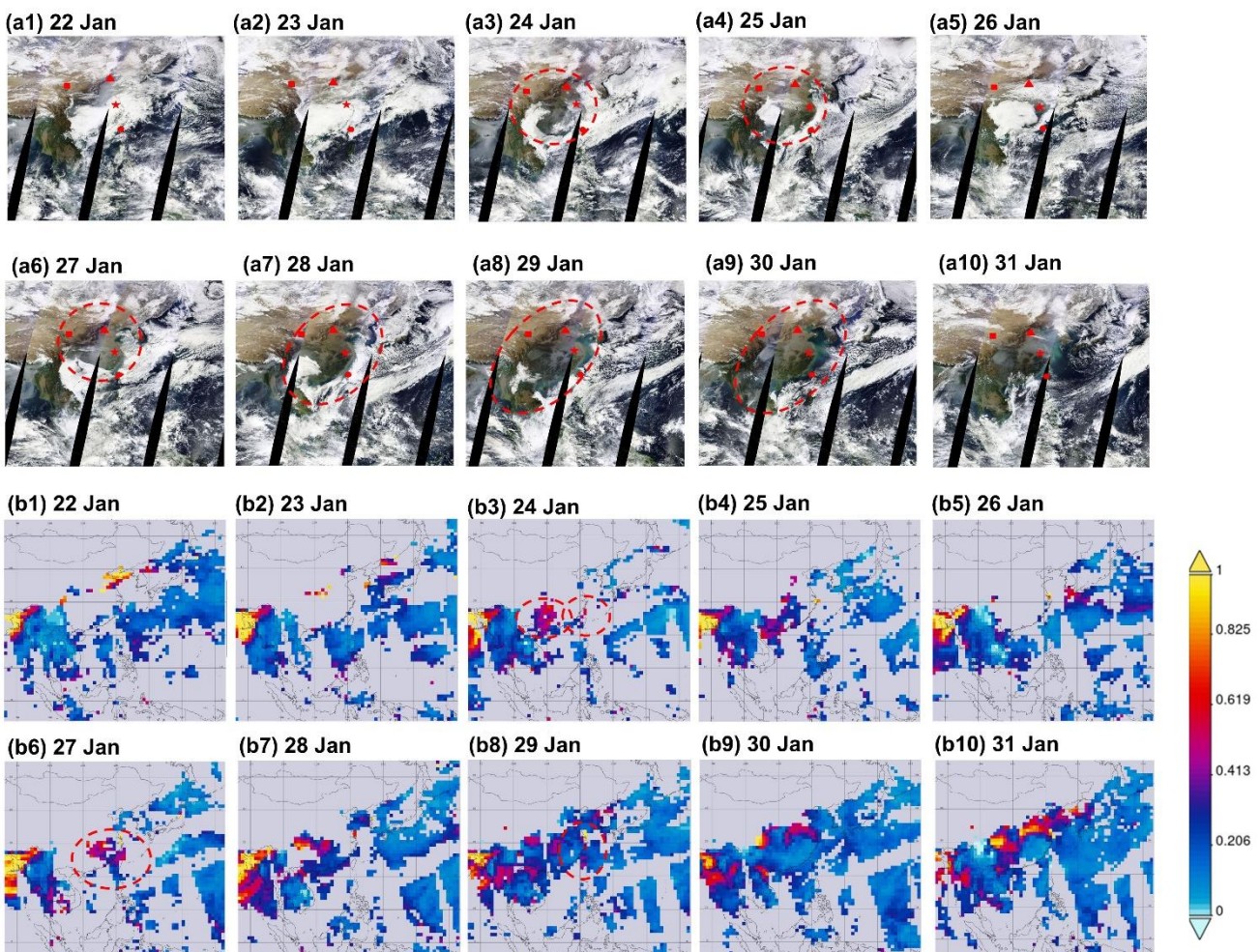


**Figure 2:** MODIS Terra images (a1-a10) and MODIS aerosol optical depth AOD at 550 nm (b1-b10) showing dust outbreak across East Asia during 22-31 January 2023. Red Rectangular, triangle, star and circle indicate Lanzhou, Beijing, Shanghai and Taiwan. The red circle with dash line indicates the dust plume.

840

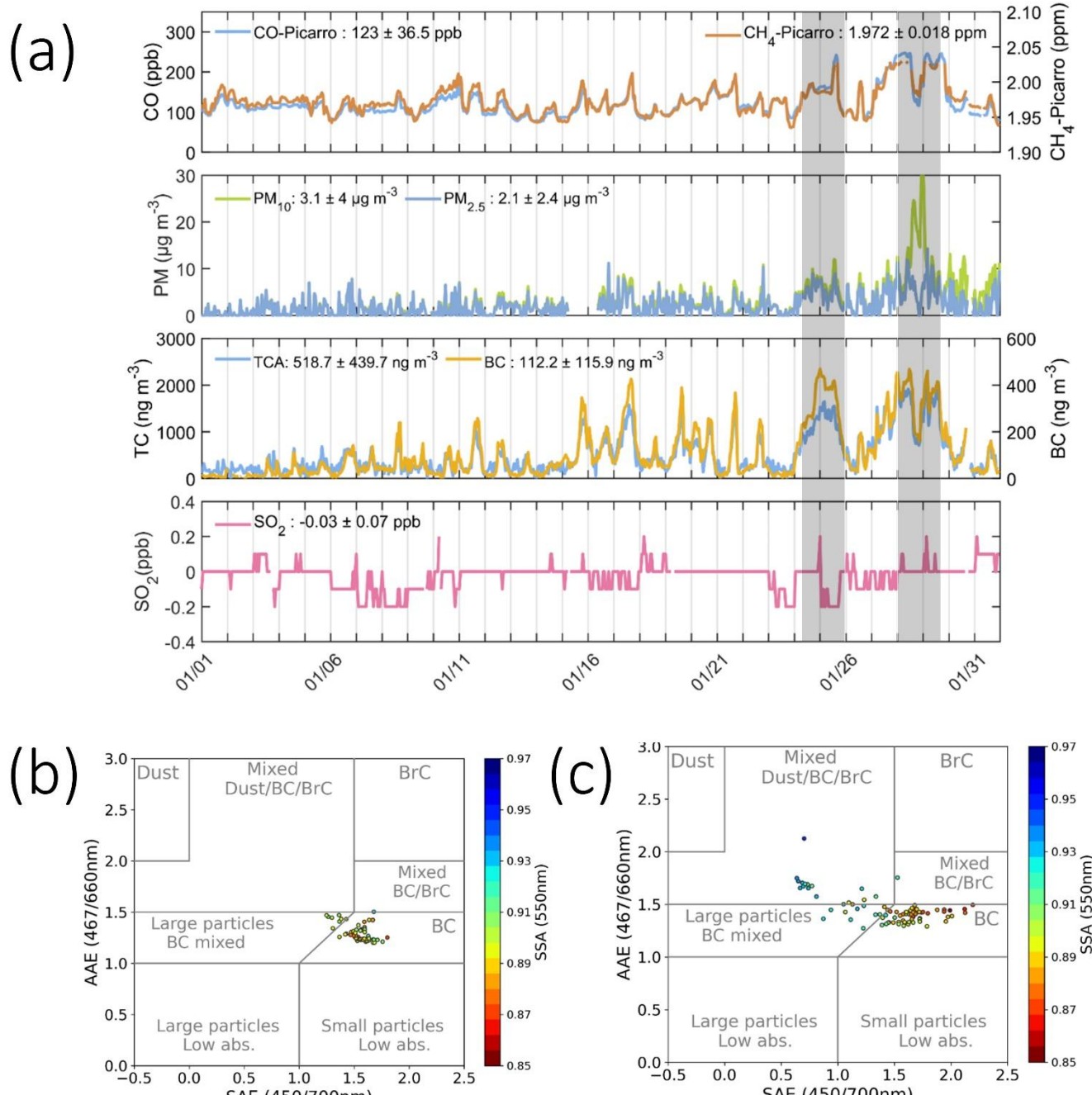

**Figure 3:** (a) Time series of observed pollutants over LABS during January 2023. The aerosol radiation properties during (b) 24-26 January and (c) 27-30 January 2023.

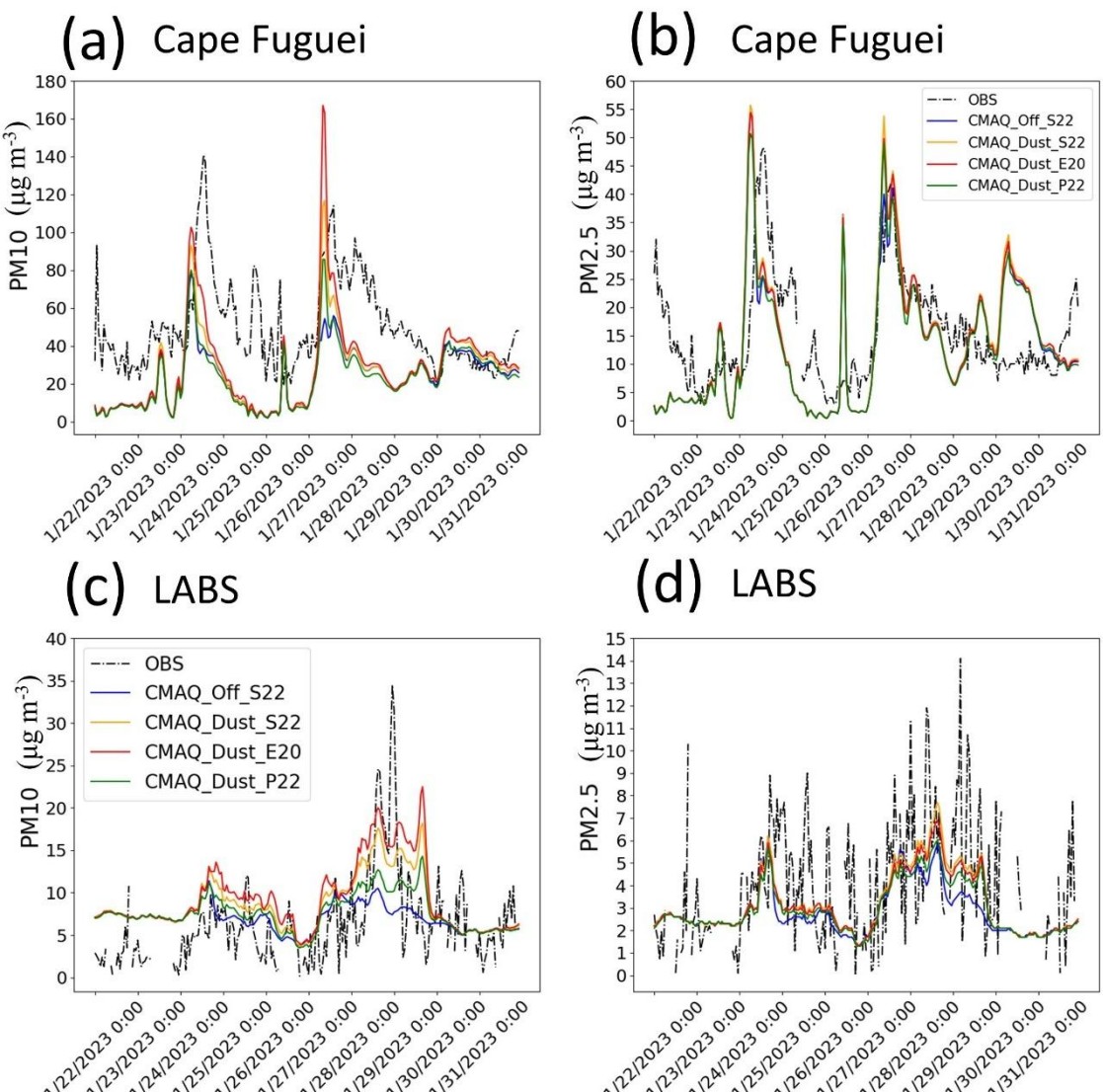

846

**Figure 4:** Time series of $PM_{10}$ (left panel) and $PM_{2.5}$ (right panel) concentrations during 22-31 January

2023 under multiple deposition schemes over the Cape Fuguei (upper panel) and LABS (lower panel),

representing the surface and high altitude, respectively.





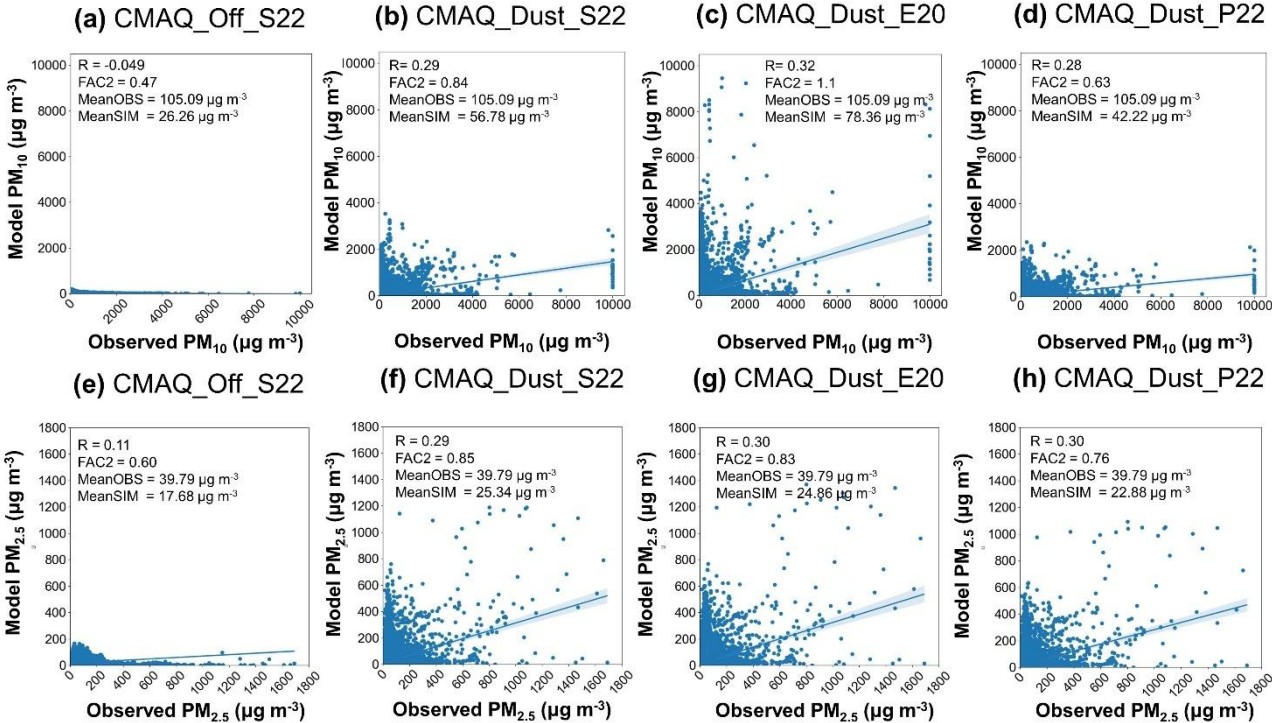

**Figure 5:** The scatter plot of the observed against modeled $PM_{10}$ (a-d) and $PM_{2.5}$ (e-h) for CMAQ_Off_S22 (a, e), CMAQ_Dust_S22 (b, f), CMAQ_Dust_E20 (c, g) and CMAQ_Dust_P22 (d, h) at the 100 sites of the mainland China on 12 March-20 April 2021 (http:// ). R is the correlation coefficient between the observation and model; FAC2 is the factor of two; MeanOBS and MeanSIM are the mean of PM from observation and model, respectively.

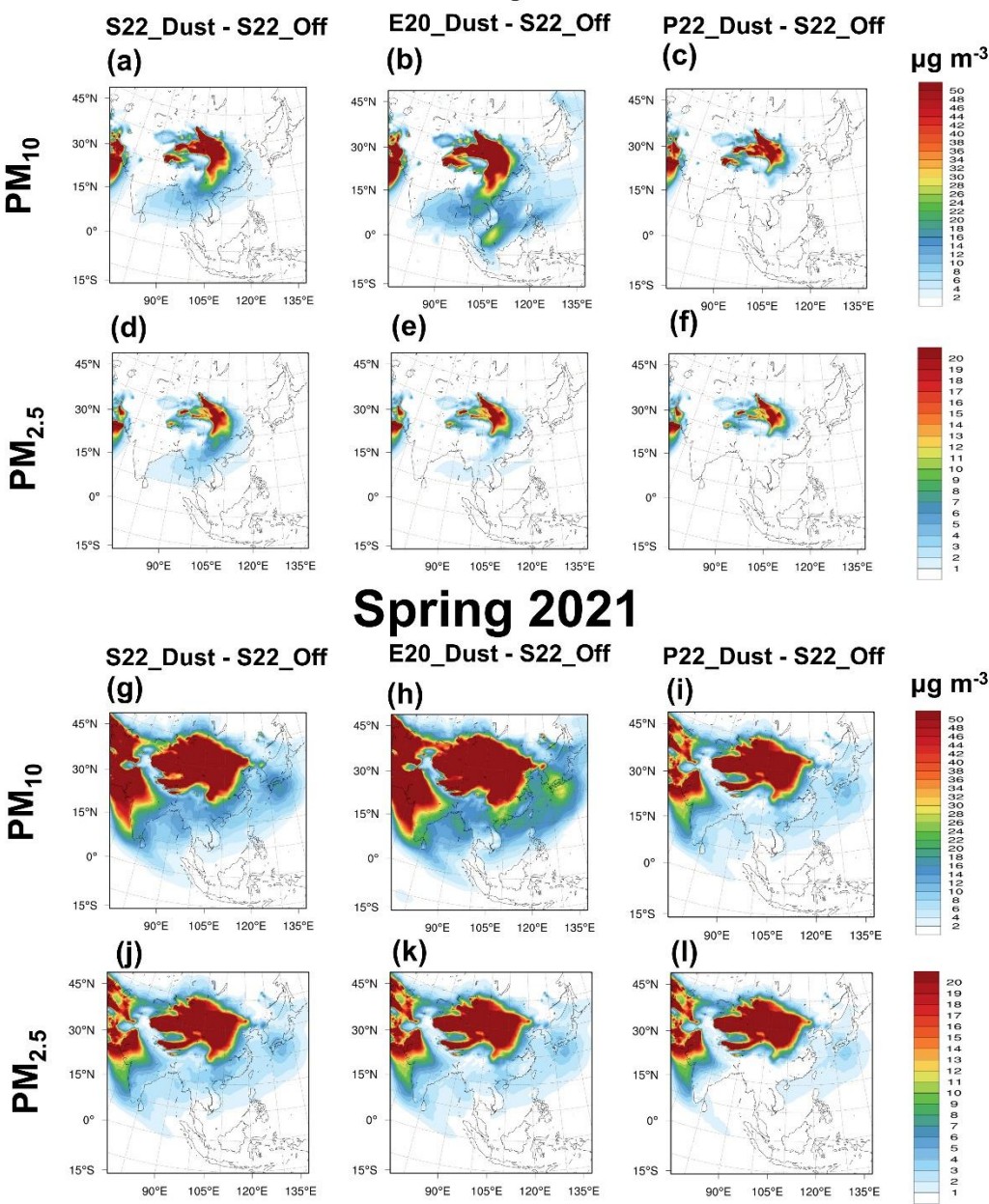

861

**Figure 6:** CMAQ estimated 10 days (January 2023) (a-f) and 40 days (Spring 2021) (g-l) averaged mean
(a, b, c, g, h, i) $PM_{10}$ and (d, e, f, j, k, l) $PM_{2.5}$ for the concentration changes using (a, d, g, j) S22, (b, e, h,
k) E20 and (c, f, i, l) P22 schemes, as relative to the CMAQ_Off_S22 scenarios.

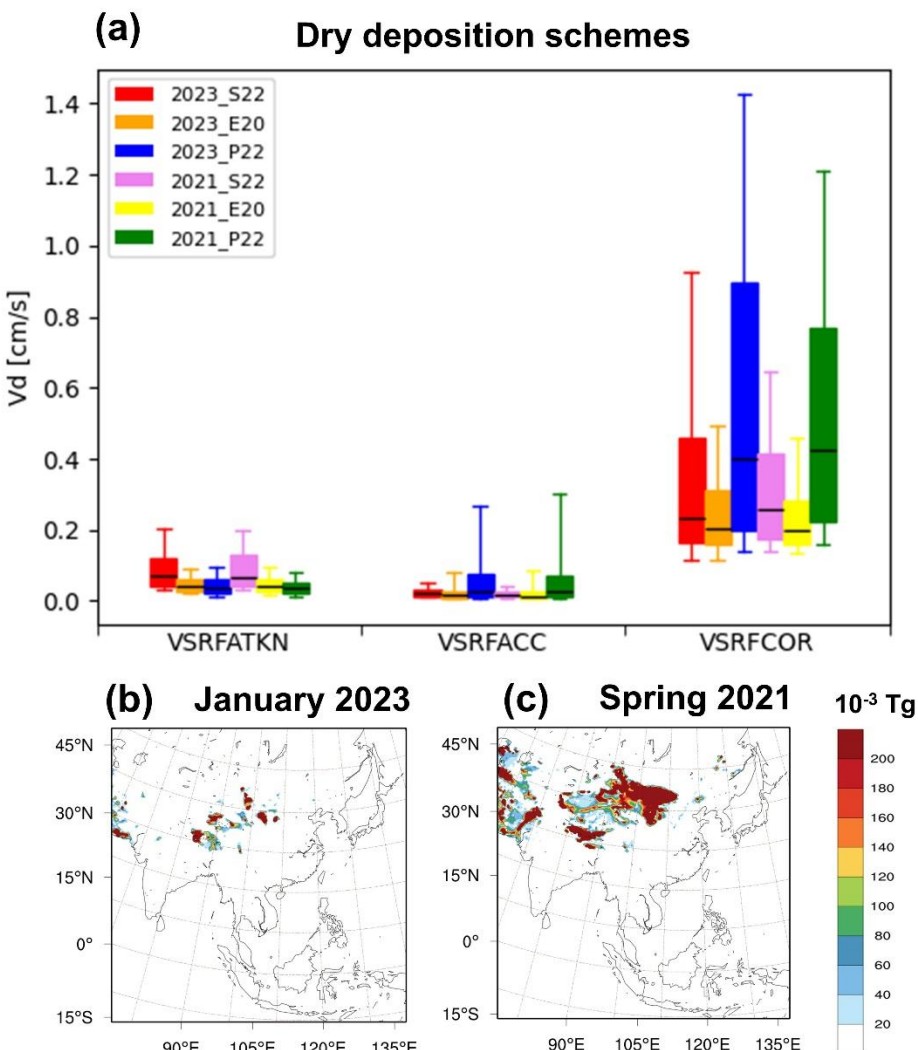

865

**Figure 7:** (a) 10-days (2023) and 40-days (2021) averaged dry $V_d$ predicted by CMAQ for the Aitken, accumulation, and coarse particle modes using the 2023_S22 (red), 2023_E20 (orange), 2023_P22 (blue), 2021_S22 (violet), 2021_E20 (yellow) and 2021_P22 (green) particle dry deposition schemes. The variability illustrated by the boxes and whiskers corresponds to spatial variability in annually averaged values throughout the CMAQ domain. The simulated total dust emission from CMAQ_Dust_E20 in (b) January 2023 and (c) Spring 2021.

872

# January 2023

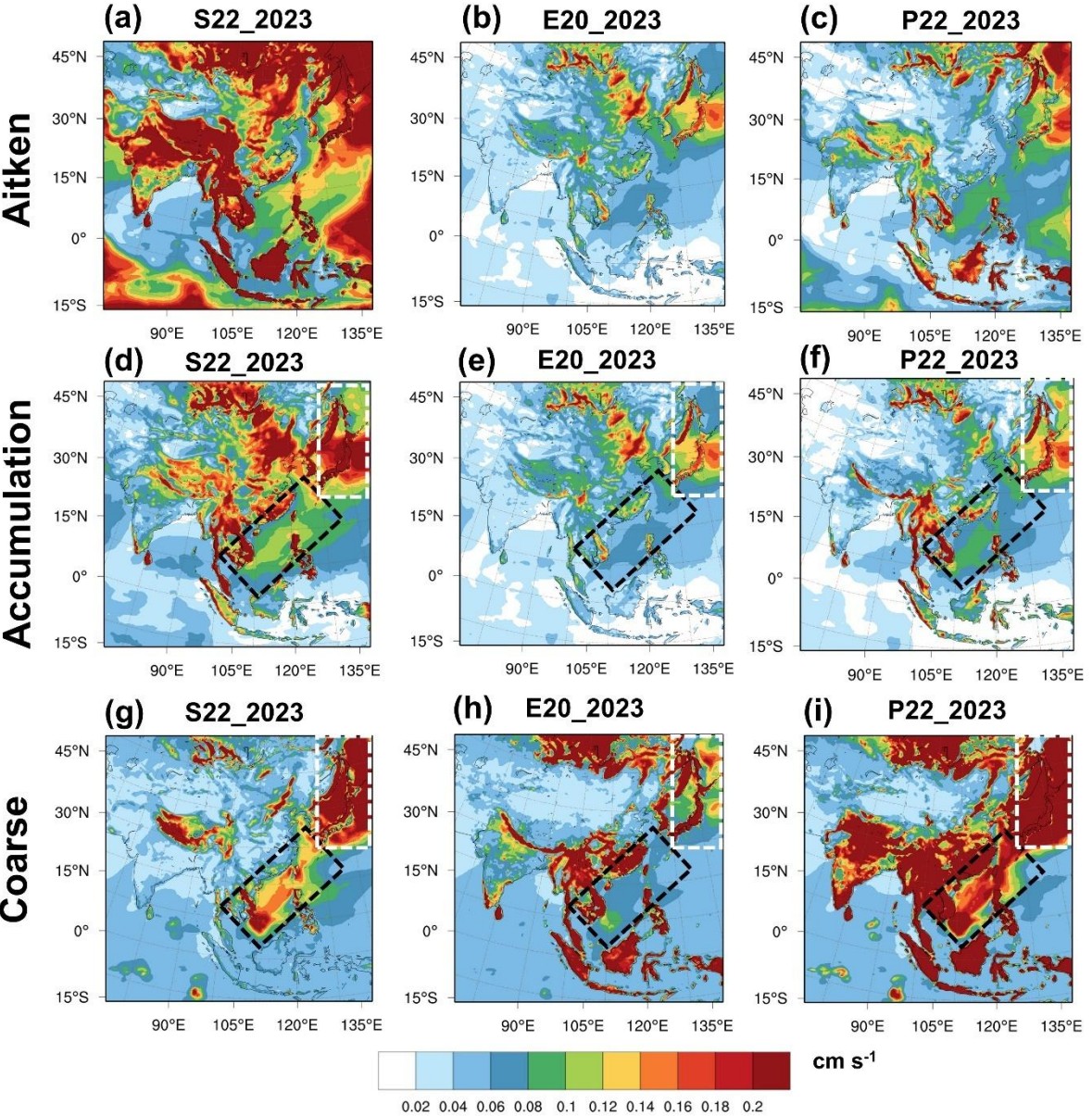

**Figure 8:** CMAQ estimated 10 days (22-31 January 2023) averaged for the (a-c) Aitken, (d-f) accumulation, and (g-i) coarse particle modes for (a, d, g) S22, (b, e, h) E20, and (c, f, i) P22 dry deposition schemes. White-dash rectangular indicates the region across northwest China; Black-dash rectangular indicates the marine boundary layer at the western Pacific.

# Spring 2021

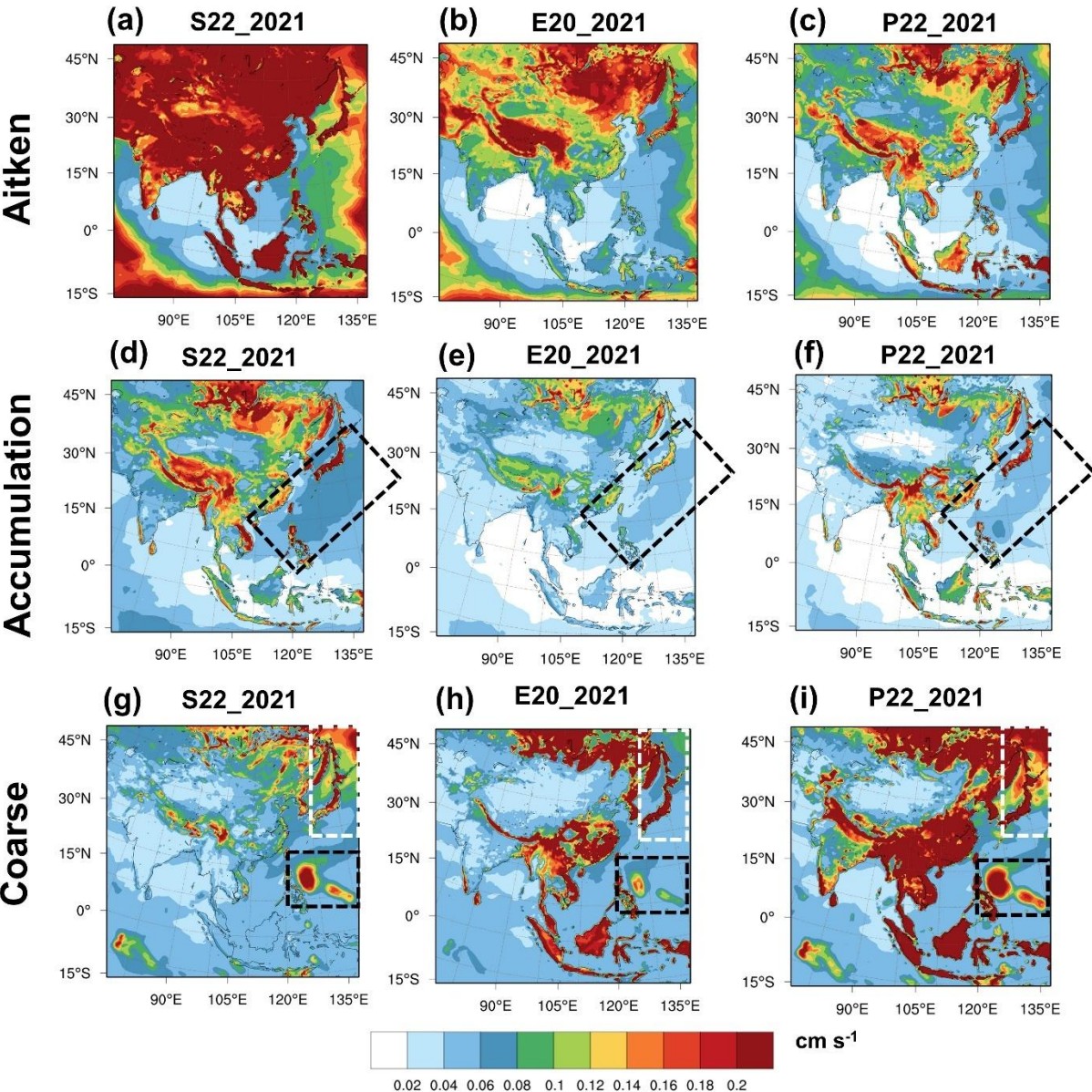

**Figure 9:** CMAQ estimated 40 days (12 Mar-20 April 2021) averaged for the (a-c) Aitken, (d-f) accumulation, and (g-i) coarse particle modes for (a, d, g) S22, (b, e, h) E20, and (c, f, i) P22 dry deposition schemes. White-dash rectangular indicates the region across northwest China; Black-dash rectangular indicates the marine boundary layer at the western Pacific.

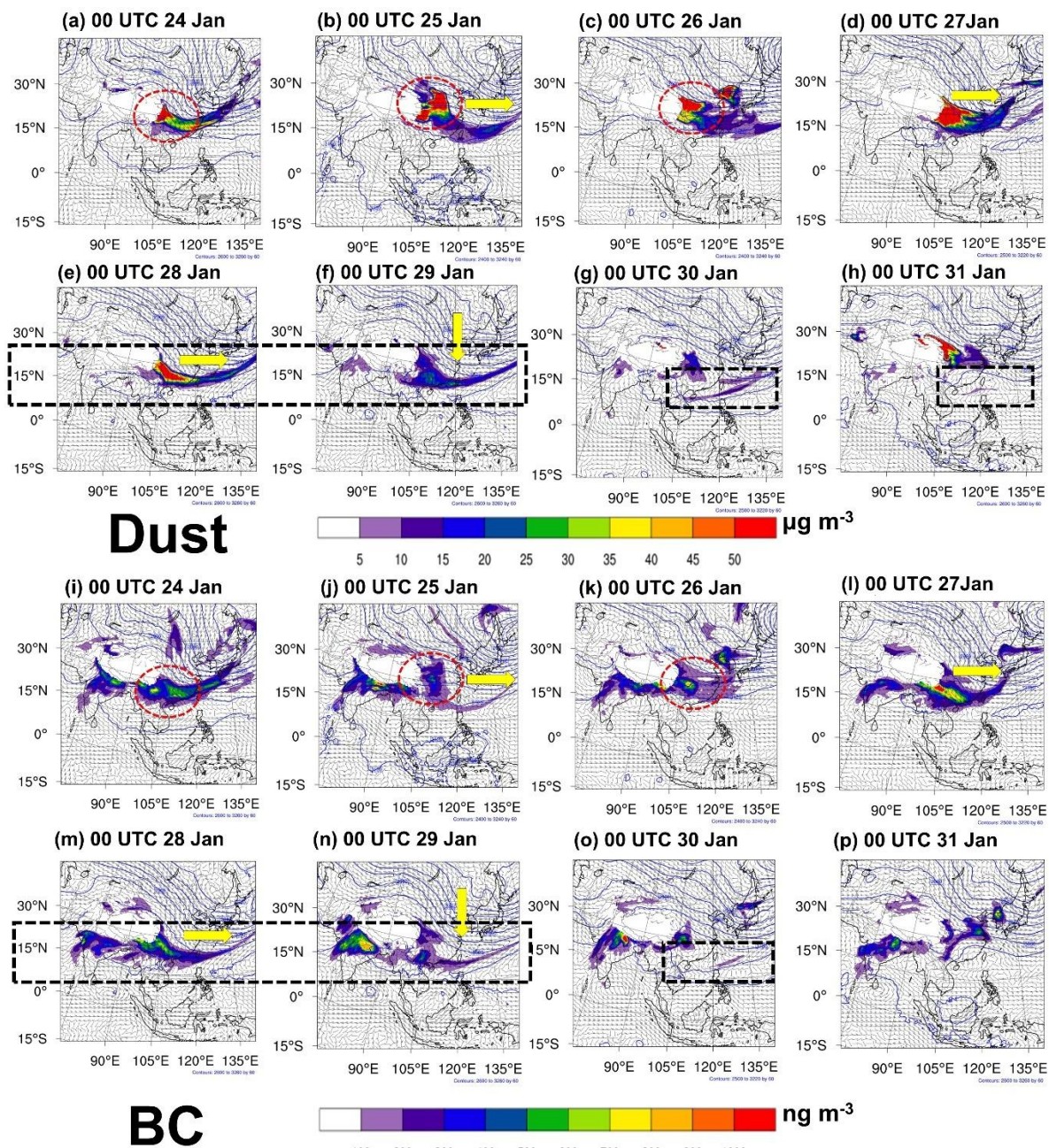

**Figure 10:** CMAQ_Dust_E20 simulated mineral dust (a-h) and BC aerosol (i-p) concentrations at the 700 hPa during 12 UTC 24-31 January 2023. The yellow arrows highlight the trough moving direction. The dash-black rectangular box highlights the aerosol belt.

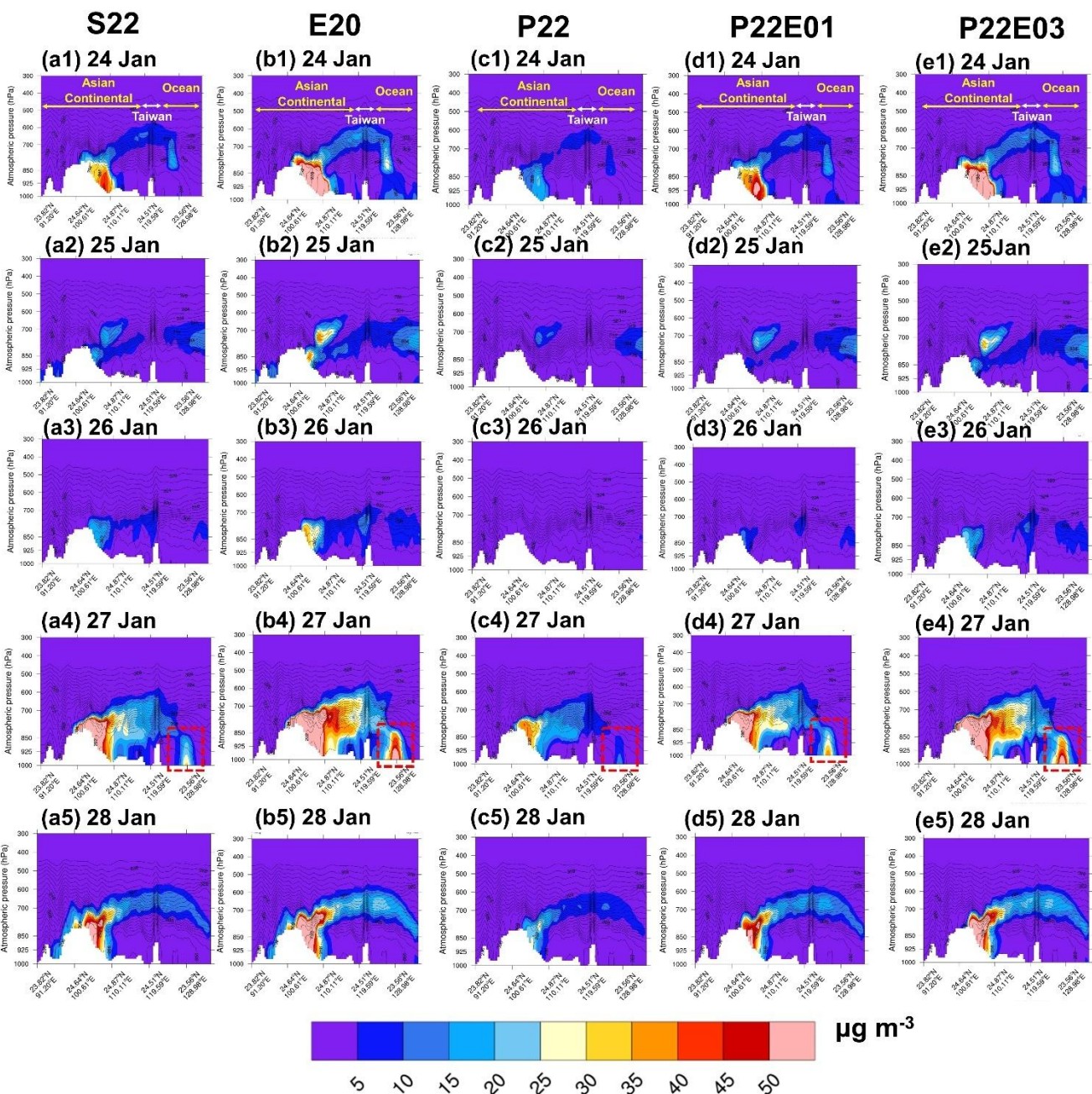


**Figure 11:** Vertical cross section of the simulated dust aerosol for the CMAQ_DUST (S22, E20, P22, P22E01 and P22E03) during 12 UTC 24-28 January 2023.



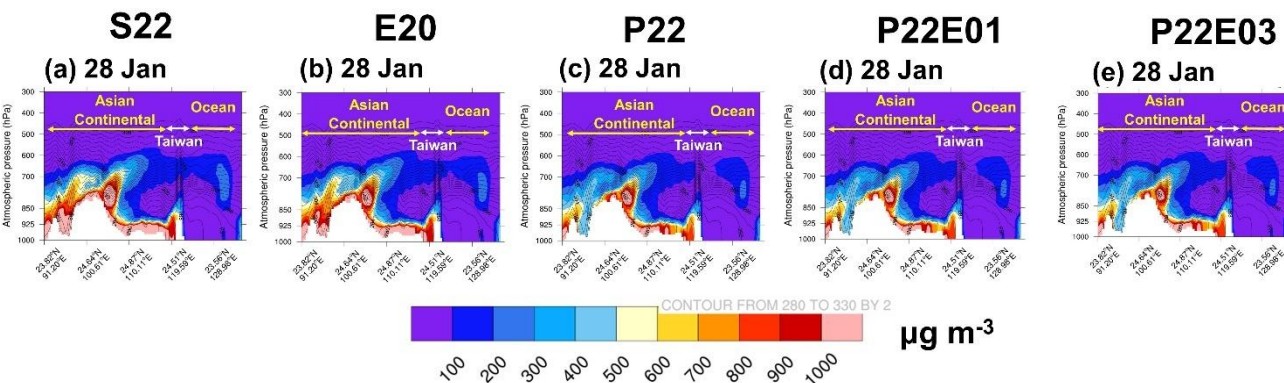


**Figure 12:** Vertical cross section of the simulated BC aerosol for the CMAQ_DUST (S22, E20, P22,
P22E01 and P22E03) during 00 UTC 28 January 2023.