# Peer review of "Modeling CMAO dry deposition treatment over Western Pacific: A distinct characteristic of mineral dust and anthropogenic aerosol"

_EGUsphere, 2024_

## Author Comment (AC1)

**Reply to the comments of RC1**

*RC1: General comment*

This paper describes a modeling study where four aerosol dry deposition schemes were used to model dust and BC. The Main point seems to be that one of the deposition schemes is best and improves the modeling of dust. This conclusion, however, is not well supported by the marginal improvement of some statistics relative to two measurements sites and MODIS AOD for brief periods during dust episodes. The problem is that the observed data is very meager and concluding that one scheme is best assumes that all other aspects of the modeling system are perfect, especially the dust emissions. Also, this a very specific application for short periods of time so there is no reason to think that these conclusions are generally relevant. Another problem with this study is that the relative performance of the four deposition schemes is not consistent with other modeling studies. In particular, the P22 scheme typically results in significantly greater deposition velocities than the E20 scheme. Also, Fig 5 shows that P22 is almost the same as PR11. This suggests that there were errors made in running these models. Table 2 says that CMAQv5.4 was used. If the STAGE option was used for dry deposition (which should be noted in the Table) a choice of S22, E20, and P22 are available. However, PR11 is not. How was this used?. Table 2 also states that the NOAH LSM was used in WRF. CMAQ needs several parameters from WRF that typically are output when using the PX LSM. When NOAH is used, default calculations for these parameters, which are important for deposition, are made in the Meteorology-Chemistry Interface Processor (MCIP). These calculations are not the same as in the LSM and will result in additional errors. There are many grammatical and other sloppy errors in the text. Far too many for me to correct.

**Response:** We greatly appreciate the referee#1 for the suggestions to improve the manuscript. All of the changes in the revised manuscript have been highlighted in ==yellow==. Corrections (blue text) with line numbers indicated in this response document refer to the revised manuscript.

Our point-by-point responses to the reviewer's comments are given below:

**General comment 1:** This paper describes a modeling study where four aerosol dry deposition schemes were used to model dust and BC. The Main point seems to be that one of the deposition schemes is best and improves the modeling of dust. This conclusion, however, is not well supported by the marginal improvement of some statistics relative to two measurements sites and MODIS AOD for brief periods during dust episodes. The problem is that the observed data is very meager and concluding that one scheme is best assumes that all other aspects of the modeling system are perfect, especially the dust emissions. Also, this a very specific application for short periods of time so there is no reason to think that these conclusions are generally relevant.

**Response:** We thank the reviewer for the comments. For the reviewer's concern, the present study focuses on the difference of the dust model improvement using the dry deposition schemes embedded in CMAQv5.4, particularly over the western Pacific region, which used Taiwan as the primary receptor. The importance of using Taiwan's observation data for CMAQ model evaluation has been highlighted. We have mentioned as "The model performance in Taiwan is paramount in our study, as the area is equipped with a substantial number of well-maintained surface observation sites, providing comprehensive coverage. The LABS station in the high-altitude subtropical western North Pacific region serves as the sole background station for monitoring transboundary pollutants. This station is crucial in our research as it provides

unique data on the long-range transport of pollutants, further underscoring the relevance of our study." **Page 3-4, Line 79-83**.

We also agree that dust emissions at the Asian continental level should be considered. We have included the model evaluation over most parts of mainland China, considering the dust source and nearby source region. 100 observation sites have been obtained from the Chinese air quality online monitoring analysis platform's website (www.aqistudy.cn/). The averaged observed $PM_{10}$ and $PM_{2.5}$ concentrations from the responding dataset were used to evaluate the model performance for the latest dust emission scheme, as well as four dry deposition schemes. Also, instead of the 10-day simulation in January 2023, we widened the simulation period by considering the multiple dust storm episodes during spring 2021 for about 40 days of simulation (12 Mar-20 Apr 2021). We added the discussion as "In addition, the hourly $PM_{10}$ and $PM_{2.5}$ of nearly 100 sites distributed over mainland China (Fig. S1), covering the period of 12 March-20 April 2021, obtained from Chinese air quality online monitoring analysis platform's website (www.aqistudy.cn/)." **Page 8, Line 188-190**.

Also, we have added the discussion as "During the spring of 2021, a series of dust storms (15 March, 27 March, and 18 April) occurred over the Gobi area, with one of the most significant dust storms in the past decade (15 March, the "3.15" dust storm hereafter) causing environmental impact over the continental (Jin et al., 2022; Gui et al., 2022; He et al., 2022; Liang et al., 2022; Tang et al., 2022). More interestingly, one of the multiple dust storm episodes reached western Pacific Ocean due to the extreme typhoon episode (Kong et al., 2024). Hence, we intend to re-emphasize the precision of various deposition schemes on the CMAQ for the recent dust storm episode over the Asian Continental highlighted by Kong et al. (2024). We evaluated the CMAQ simulations with the different dry deposition schemes for the 40-day sensitivity test on 12 March-20 April 2021 against measured $PM_{10}$ and $PM_{2.5}$ concentrations across the observation sites in mainland China (Table 4). The observation sites used for the model comparison are marked in Fig. S1. Generally, the evaluation results for Taiwan and mainland China were consistent. During the 40 days of Spring 2021, the CMAQ $PM_{10}$ of NMB was the highest for Off_PR11 (NMB = -79.19 %), followed by Dust_PR11 (-60.53 %). The latest inline dust emission scheme embedded with E20 dry deposition scheme for $PM_{10}$ was well performed by NMB of -25.43 %, compared to the Dust_S22 (-45.97 %) and Dust_P22 (-59.82 %). For the $PM_{2.5}$ simulation, Dust_PR11 has been improved from Dust_Off, and Dust_S22 was slightly better than Dust_E20.

Figure 5 shows the scatter plot of simulated and observed PM across mainland China. The correlation coefficient (R), a factor of two (FAC2), and the mean observed and simulated PM are marked in Figure 5. The modeled $PM_{10}$ without the dust scheme had the lowest correlation, followed by Dust_PR11. Among all of these simulations, Dust_E20 performed the best correlation (R > 0.3) compared to Dust_PR11, Dust_S22 and Dust_P22. However, for $PM_{2.5}$, the correlation between the model and measured values was similar for all the dry deposition schemes. The statistical index of FAC2 was used in the present work since either low or high outliers less influence it (Chan and Hanna, 2004). The dataset is reliable for FAC2 values between 0.5 and 2.0, with the ideal model of 1.0. The simulated $PM_{10}$ by E20 performed well, with a nearly perfect value of 1.1. Meanwhile, the $PM_{2.5}$ by S22 simulation was slightly better than E20 but much better than the other experiments." **Page 11, Line 259-284**.

**Table 4.** CMAQ evaluation for $PM_{10}$ and $PM_{2.5}$ against the averaged 100 observation sites across mainland China (Fig. S1) and AOD against MODIS daily observation near the dust source region (above 30°N) with Normalized Mean Bias (NMB) under the multiple simulation

scenarios (Fig. S3). Spring 2021, 3.15, 3.27, and 4.18 represent the evaluation period by 12 March-20 April 2021, 14-16 March 2021, 26-28 March 2021, and 17-19 April 2021, respectively.

| Parameters | Period | CMAQ-M3DRY | | CMAQ-STAGE | | |
|---|---|---|---|---|---|---|
| | | Off_PR11 | Dust_PR11 | Dust_E20 | Dust_S22 | Dust_P22 |
| $PM_{10}$ | Spring 2021 | -79.15 | -60.53 | -25.43 | -45.97 | -59.82 |
| $PM_{2.5}$ | Spring 2021 | -60.94 | -44.84 | -37.50 | -36.29 | -42.47 |
| AOD | 3.15 | -81.92 | -49.54 | -38.97 | -46.41 | -48.45 |
| | 3.27 | -75.10 | -46.12 | -36.39 | -41.84 | -44.52 |
| | 4.18 | -55.88 | -16.49 | -3.20 | -7.83 | -14.52 |
| | Mean AOD | -70.97 | -37.38 | -26.19 | -32.03 | -35.83 |

[Figure]

**Figure 5:** The scatter plot of the observed against modeled $PM_{10}$ (a-e) and $PM_{2.5}$ (f-j) for CMAQ_Off_PR11 (a, f), CMAQ_Dust_PR11 (b, g), CMAQ_Dust_E20 (c, h), CMAQ_Dust_S22 (d, i) and CMAQ_Dust_P22 (e, j), at the 100 sites of the mainland China on 12 March-20 April 2021 (http://www.aqistudy.cn/). R is the correlation coefficient between the observation and model; FAC2 is the factor of two; MeanOBS and MeanSIM are the mean of PM from observation and model, respectively.

[Figure]

**Figure S1:** The location of monitoring sites over mainland China used for model evaluation (http://www.aqistudy.cn/).

**General comment 2:** Another problem with this study is that the relative performance of the four deposition schemes is not consistent with other modeling studies. In particular, the P22 scheme typically results in significantly greater deposition velocities than the E20 scheme. Also, Fig 5 shows that P22 is almost the same as PR11. This suggests that there were errors made in running these models.

**Response:**
We agree with the reviewer regarding the P22 and E20. The greater deposition velocity is by P22 than E20, particularly in the coarse mode. The STAGE and M3DRY under CMAQv5.4 have been examined over Continental U.S.A. (CONUS) during July 2016 (USEPA, 2024). The difference in modeled particulate matter between PR11 and P22 dry deposition was within 5 %, and the time series trend was similar. To justify the present model result of PR11 and P22, we purposely replot the corresponding PM10 and PM25 for a clear comparison (Fig. S5). Moreover, the STAGE by E20 simulated a higher $PM_{2.5}$ compared to STAGE P22 (USEPA, 2024). This shows that the model testing results over the CONUS were consistent with the East Asia domain demonstrated in the present study.

[Figure]

**Figure S6:** Time series of $PM_{10}$ (left panel) and $PM_{2.5}$ (right panel) concentrations during 22-31 January 2023 under CMAQ_Dust_PR11 and CMAQ_Dust_P22 simulations over the Cape Fuguei (upper panel) and LABS (lower panel), representing the surface and high altitude, respectively.

Also, we have updated the STAGE dry deposition description as "Moreover, Surface Tiled Aerosol and Gaseous Exchange (STAGE) deposition has been implemented within the CMAQv5.3, where estimated fluxes from sub-grid cell fractional land-use values, aggregates the fluxes to the model grid cell and unifies the bidirectional and unidirectional deposition schemes using the resistance framework (Massad et al., 2010; Nemitz et al., 2001). The updated STAGE version in CMAQv5.4 could aggregate the grid-scale values that match the grid-scale values from most kinds of Land Surface Model of WRF (Hogrefe et al., 2023)." **Page 6, Line 146-152**.

References:
U.S. Environmental Protection Agency: https://github.com/USEPA/CMAQ/wiki/CMAQ-Release-Notes:-Dry-Deposition-Air-Surface-Exchange:-Surface-Tiled-Aerosol-and-Gaseous-Exchange-(STAGE), last access 15 October 2024.

Hogrefe, C., Bash, J. O., Pleim, J. E., Schwede, D. B., Gilliam, R. C., Foley, K. M., Appel, K. W., and Mathur, R.: An analysis of CMAQ gas-phase dry deposition over North America through grid-scale and land-use-specific diagnostics in the context of AQMEII4, Atmos. Chem. Phys., 23, 8119–8147, https://doi.org/10.5194/acp-23-8119-2023, 2023.

**General comment 3:** Table 2 says that CMAQv5.4 was used. If the STAGE option was used for dry deposition (which should be noted in the Table) a choice of S22, E20, and P22 are available. However, PR11 is not. How was this used?.

**Response:** We have included the dry deposition options in Table 1. Also, the corresponding changes have been used to modify the scenario description in Table 2. We change the tables as below:

**Table 1.** Model settings.

| Model setting | Descriptions |
| --- | --- |
| Period | 12 March-20 April 2021 and 22-31 January 2023 |
| Domain | d01, d02, and d03 with 45 KM, 15 KM, and 5 KM of the resolutions, respectively |
| Boundary condition | NCEP FNL lateral boundary condition |
| Surface and land surface model | NOAH |
| Numerical weather model | WRF v40, including grid and observation nudging at d01. |
| Chemical transport model | CMAQ v5.4 |
| Gas-phase chemistry and aerosol mechanism | CB06e51 + AE7 |
| Emission Inventory | MICS-ASIA III emission in 2023, adjusted from the emission in 2017 (Zhang et al., 2018) based on the OMI-NO$_X$ satellite (Huang et al., 2021). |
| Online dust treatment | The windblown dust treatment suggested by Kong et al. (2024). |
| Dry deposition option | M3DRY (PR11) and STAGE (E20, S22 and P22). |

**Table 2.** Simulation scenarios used in this present study.

| Scenarios | Descriptions |
| --- | --- |
| CMAQ_Off_PR11 | Without in-line dust calculation, with the M3DRY dry deposition algorithm by Pleim and Ran (2011). |
| CMAQ_Dust_PR11 | Implement the latest refined dust treatment proposed by Kong et al. (2024), with the M3DRY dry deposition algorithm by Pleim and Ran (2011). |
| CMAQ_Dust_E20 | Same as CMAQ_Dust_PR11, but with the STAGE dry deposition algorithm by Emerson et al. (2020). |
| CMAQ_Dust_S22 | Same as CMAQ_Dust_PR11, but with the STAGE dry deposition algorithm by Shu et al. (2022). |
| CMAQ_Dust_P22 | Same as CMAQ_Dust_PR11, but with the STAGE dry deposition algorithm by Pleim et al. (2022). |

**General comment 4:** Table 2 also states that the NOAH LSM was used in WRF. CMAQ needs several parameters from WRF that typically are output when using the PX LSM. When NOAH is used, default calculations for these parameters, which are important for deposition, are made in the Meteorology-Chemistry Interface Processor (MCIP). These calculations are not the same as in the LSM and will result in additional errors.

**Response:** We greatly appreciate the reviewer's comments. The reason for using PXLSM is to get the look-up table directly from WRF PXLSM, which would be readable by the MCIP calculation needed for the deposition algorithm. However, under the STAGE option, the

mapping by source code in ASX_DATA_MOD.F could be set up to simulate most kinds of LSM, including NOAH and CLM. With NOAH, the parameterizations, including Z0, VEG, and LAI, were directly taken from the WRF look-up table. In addition, the NOAH-LSM contains additional LU importance in simulating dust deserts such as playa and white sand, which is not included in PXLSM. So, applying NOAH-LSM over East Asia, which experienced frequent dust episodes, could be more representative. However, we do realize that due to the difference in LU categories and soil type of both PXLSM and NOAH, the impact on aerosol (PM10/PM25) emission and deposition can vary, and we shall propose this idea as our future model testing.

**General comment 5:** There are many grammatical and other sloppy errors in the text. Far too many for me to correct.

**Response:** We thank the reviewer for the suggestion. The grammar of the manuscript has been proofread by Grammarly (version Premium) online text editor.

**RC1: Specific comments and responses**

**Comment 1:** Table 1 has many errors. For example, the equations for PR11 are all for gasses not aerosols. Please remove Table 1.

**Response:** We thank the reviewer for the suggestion. Table 1 has been removed.

**Comment 2:** Line 17. P20 should be P22.

**Response:** We have revised the term P20 to P22. **Page 1, Line 20**.

**Comment 3:** L19. This sentence implies that dry deposition directly affects dust emissions.

**Response:** The sentence has been revised. We corrected the sentence as "The result showed that the dry deposition parameterization could significantly improve the CMAQ in simulating $PM_{10}$ and $PM_{2.5}$ concentrations." **Page 1, Line 17-18**.

**Comment 4:** L30. Sentence does not make sense.

**Response:** The sentence emphasized the LABS measurement located at 2862 m.s.l. showed the mixing of dust and black carbon from 22-31 January 2023. We corrected the sentence as "On 22-31 January 2023, the *in-situ* measurement of the upper level observed the coexistence of natural dust and anthropogenic aerosol." **Page 1, Line 29-30**.

**Comment 5:** L32-33. This sentence "resolving the uncertainty of the CMAQ dust emission treatment" is a gross overstatement.

**Response:** Agree. We have revised the sentence as "We proposed implementing the E20 dry deposition approach, narrowing the uncertainty of the CMAQ dust emission treatment." **Page 1, Line 31-33**.

**Comment 6:** L38. Does not make sense.

**Response:** The sentence has been revised. We modified the sentence as "Among these, particle dry deposition is a crucial aerosol removal process and an important sink for particles in the model." **Page 1, Line 36-38**.

**Comment 7:** L51. Again, a gross overstatement: "Emerson et al. (2020) has resolved the problem."

**Response:** Agree. We have revised the sentence as "The latest dry deposition scheme revision by Emerson et al. (2020) has reduced the uncertainty, marking a significant step forward in our quest for more accurate air quality modeling." **Page 2, Line 50-52**.

**Comment 8:** L64. What "boarder"?

**Response:** We have revised the sentence as "The surface fine particle concentrations can vary up to 5-15 %, and the particle dry deposition has more than 200 % discrepancy due to the different dry deposition schemes. (Saylor et al., 2019)." **Page 3, Line 62-64**.

**Comment 9:** L70-72. Does not make sense.

**Response:** The statement mentioned the simulated $PM_{10}$ underestimation caused by the uncertainty of the deposition mechanism. We changed the sentence as "Besides the model bias on $PM_{2.5}$, the simulation of $PM_{10}$ has been underestimated due to the uncertainty of the deposition mechanism, particularly over the western Pacific." **Page 3, Line 69-70**.

**Comment 10:** L92-93. Fix notation.

**Response:** The notation has been fixed. We modified the notation as "
$$u_{*,t} = u_{*,to} f_m f_r \tag{1}$$
Where $u_{*,to}$ is the ideal threshold friction velocity, while $f_m$ and $f_r$ are the correction factors of soil moisture and surface roughness, respectively." **Page 5, Line 113-115**.

**Comment 11:** L100. Bulb?

**Response:** The word has been corrected as "…and bulk soil density…". **Page 5, Line 122.**

**Comment 12:** L115-118. This is very sloppy. Please fix Vs and Vg

**Response:** The physical formulation and notation have been corrected as below: "

$$V_d = V_s + \frac{1}{R_a + R_s} \tag{3}$$

where $V_s$ is the gravitation settling velocity, $R_a$ is the resistivity aerodynamic and $R_s$ is the surface resistivity. The $V_s$ is calculated according to Stokes's Law as:

$$V_s = \frac{p_p \, D_p^2 \, g C_c}{18 \eta} \tag{4}$$

where, $p_p$ is the density of the particle; $D_p$ is the diameter of the particle; g is gravitational acceleration; $C_c$ is the Cunningham correction factor for small particles; and, η is the dynamic viscosity of air." **Page 6, Line 137-142**.

**Comment 13:** L122-123. This is not true: "Dry deposition is based on gravitational settling velocity (Vg), which is the function of aerodynamic and surface resistance."

**Response:** Agree. The statement has been removed.

**Comment 14:** L126. Should note that STAGE is one of two options, the other being M3Dry.

**Response:** Agree. The methodology has been revised as "CMAQ is embedded with M3Dry dry deposition calculation that implements the scheme of Pleim and Ran (2011), which is based on Slinn (1982). As noted by Pleim and Ran (2011), chemical surface flux modeling has become an essential process in the air quality model. For instance, the linkages of ambient concentration levels to the deposition of $SO_x$ and $NO_x$. Moreover, Surface Tiled Aerosol and Gaseous Exchange (STAGE) deposition has been implemented within the CMAQv5.3, where estimated fluxes from sub-grid cell fractional land-use values, aggregate the fluxes to the model grid cell and unifies the bidirectional and unidirectional deposition schemes using the resistance framework (Massad et al., 2010; Nemitz et al., 2001). The updated STAGE version in CMAQv5.4 could aggregate the grid-scale values that match the grid-scale values from most kind of Land Surface Model of WRF (Hogrefe et al., 2023)." **Page 6, Line 143-152**.

**Comment 15:** L137. Here, and many other places, abbreviations such as PSEA are used without defining. Also, SDS, WPO.

**Response:** The abbreviations have been defined as "…peninsular Southeast Asia (PSEA)…". **Page 7, Line 159**.

The abbreviation SDS is removed and replaced with the full defining terms. We modified it to "Super Dust Storm." **Page 11, Line 289**.

The abbreviation WPO is removed and replaced with the full defining terms. We modified it to "western Pacific Ocean." **Page 9, Line 210-211**.

**Comment 16:** L139. What are the chemical LBCs?

**Response:** The chemical LBC was generated by a time-invariant set of predefined, vertical concentration profiles. For nested simulations, the dynamic boundary conditions are extracted from CCTM output from a coarse-grid simulation (USEPA, 2010).

Reference:
U.S. Environmental Protection Agency: Operational Guidance for the Community Multiscale Air Quality (CMAQ) Modeling System Version 4.7.1, https://www.cmascenter.org/cmaq/documentation/4.7.1/Operational_Guidance_Document.pdf, 2010.

**Comment 17:** L156. CMAQ_Dust_PR11 is repeated.

**Response:** We thank the reviewer for the comment. The additional "CMAQ_Dust_PR11" is removed.

**Comment 18:** L177.  Don't see dust claw.

**Response:** The statement has been revised. We modified the sentence as "The satellite image showed dust induced by a high-pressure system on 24-25 January (Fig. 2a3, 2a4). The next day, the same region was covered by a thick cloud, and dust was again widely distributed from 27-30 January 2023." **Page 8, Line 202-204**.

**Comment 19:** L199.  Table 3 should be 4.

**Response:** Table 3 remains. **Page 9, Line 224**.

**Comment 20:** L209.  Numbers are reversed.

**Response:** We thank the reviewer for the comment. The numbers in the text are correct as Table 3 has been modified between the columns of Dust_S22 and Dust_P22. We have corrected the table as below:

**Table 3.** Statistical evaluation for $PM_{10}$ and $PM_{2.5}$ concentrations during 22-31 January 2023 for Cape Fuguei under the multiple simulation scenarios.

| | Benchmark | CMAQ-M3DRY | | CMAQ-STAGE | | |
|---|---|---|---|---|---|---|
| | | Off PR11 | Dust_PR11 | Dust_E20 | Dust_S22 | Dust_P22 |
| **$PM_{10}$** | | | | | | |
| MeanObs | | 49.97 | 49.97 | 49.97 | 49.97 | 49.97 |
| MeanMod | | 21.19 | 22.97 | 29.04 | 26.48 | 23.04 |
| NMSE | | 0.82 | 0.71 | 0.49 | 0.56 | 0.71 |
| NMB | ± 85% | -57.59 | -54.05 | -41.90 | -47.01 | -53.90 |
| Corr | > 0.35 | 0.41 | 0.44 | 0.52 | 0.46 | 0.42 |
| NMBF | | -1.36 | -1.18 | -0.72 | -0.89 | -1.17 |
| | | | | | | |
| **$PM_{2.5}$** | | | | | | |
| MeanObs | | 15.52 | 15.52 | 15.52 | 15.52 | 15.52 |
| MeanMod | | 12.48 | 12.95 | 13.86 | 14.15 | 13.16 |
| NMSE | | 0.31 | 0.29 | 0.29 | 0.30 | 0.31 |
| NMB | ± 85% | -19.55 | -16.53 | -10.65 | -8.84 | -15.22 |
| Corr | > 0.35 | 0.52 | 0.55 | 0.53 | 0.53 | 0.52 |
| NMBF | | -0.24 | -0.20 | -0.12 | -0.10 | -0.18 |

**Comment 21:** Fig 4.  Hard to tell the different model runs apart especially for the PM25.  It would help to expand the scale on the PM25 plots.

**Response:** The scale on the PM2.5 plots in Fig.4 has been expanded. We modified the figure as:

[Figure]

**Figure 4:** Time series of $PM_{10}$ (left panel) and $PM_{2.5}$ (right panel) concentrations during 22-31 January 2023 under multiple deposition schemes over the Cape Fuguei (upper panel) and LABS (lower panel), representing the surface and high altitude, respectively.

**Comment 22:** L224. Where is "the north peninsula of Southeast Asia"?

**Response:** The term has been revised. We have modified the term as "northern PSEA".
**Page 10, Line 250.**

**Comment 23:** L251. There is something wrong here. P22 and PR11 should be very different.

**Response:** Please refer to the response of **General Comment 2**.

**Comment 24:** L275-276. These results seem contrary to other modeling studies where P22 generally has greater much Vd for Accumulation mode than E20. Maybe put these numbers in a table.

**Response:** Agree. The statement in the text was referring to the median of deposition velocity, which indicates that S22 was the highest median value among all. For the deposition velocity

at the 75th percentile, 75 % of the $V_d$ for P22 was higher than E20. We have modified the sentence as "As shown in the figure, the median of E20, S22, and P22 increased the deposition velocity of the Aitken (accumulation) modes particle as compared to PR11 by 22.56 (11.32) %, 117.76 (86.43) % and 2.5 (7.52) % respectively." **Page 13, Line 318-320**.

Also, the $V_d$ values of the four dry deposition schemes has been included in supplementary. We added the Table S1 as below:

Table S1: $V_d$ percentiles (cm s$^{-1}$) of atiken, accumulation and coarse particle modes by the four dry deposition schemes.

| Dry deposition schemes | Percentiles | Aitken | Accumulation | Coarse |
|---|---|---|---|---|
| PR11 | 25th | 0.033 | 0.020 | 0.046 |
|  | 50th | 0.056 | 0.036 | 0.059 |
|  | 75th | 0.086 | 0.066 | 0.186 |
| E20 | 25th | 0.043 | 0.025 | 0.046 |
|  | 50th | 0.068 | 0.040 | 0.053 |
|  | 75th | 0.102 | 0.066 | 0.092 |
| S22 | 25th | 0.076 | 0.041 | 0.043 |
|  | 50th | 0.122 | 0.068 | 0.052 |
|  | 75th | 0.201 | 0.114 | 0.077 |
| P22 | 25th | 0.034 | 0.021 | 0.048 |
|  | 50th | 0.057 | 0.039 | 0.067 |
|  | 75th | 0.089 | 0.073 | 0.208 |

**Comment 25:** L286 (Fig.7).  Why not show dry deposition velocity for each model rather that difference from PR11?

**Response:** We thank the reviewer for the question. The primary goal of our study is to enhance the performance of the latest dust model by implementing STAGE deposition schemes in contrast to the M3DRY scheme. Simply showing the difference from PR11 is most significant as it provides a clear overview of how the model has been enhanced, thereby advancing our understanding of dust modeling behavior. The dry deposition velocity for each model is also displayed at the supplementary document. We added Fig. S4 as:

[Figure]

**Figure S4:** CMAQ estimated 10 days (22-31 January 2023) averaged for the (a-d) Aitken, (e-h) accumulation, and (i-l) coarse particle modes for (a, e, i) PR11, (b, f, j) E20, (c, g, k) S22 and (d, h, l) P22 dry deposition schemes

**Comment 26:** L300. Elemental

**Response:** The term has been revised. We modified the sentence as "Black carbon, often known as elemental carbon, released from the biofuels, fossil fuels and biomass burning, has been proven to impact the radiative budget and regional climate". **Page 14, Line 346-347**.

**Comment 27:** L330. Trans what boundary?

**Response:** The term has been modified as "… the long-range transport of modeled black carbon …". **Page 15, Line 376**.

**Comment 28:** L350-351. The modeled BC in Fig 11h seems to end at Taiwan. Also, there is no Fig 11i.

**Response:** Fig. 11h is replaced by Fig. 12d; Fig. 11i has been removed and changed as Fig. 12e. We fixed the figure number as "As shown in Fig. 12d, the modeled black carbon was found distributed at the western Pacific Ocean. In Fig. 12e, a clear black carbon dome was distributed along 700 hPa, showing a similar pattern as dust." **Page 15-16, Line 396-399**.

**Comment 29:** L353. This phrase makes no sense: "as the coarse particles could comprise of fine particles".

**Response:** The phrase has been removed. We change the sentence as "This simulation proposes the consistency of the "double dome" mechanism of Asian dust and biomass burning episodes" **Page 16, Line 398-399**.

**Comment 30:** L373. "vastly" is again an overstatement.

**Response:** The term has been removed. We change the sentence as "…surface $PM_{10}$ has been improved by ..." **Page 16, Line 416**.

---

## Author Comment (AC2)

**Reply to the comments of RC2**

*RC2: General comment*

This study describes an improvement to the CMAQ dry deposition over Western Pacific using various dry deposition schemes. It also addressed the performance of a model-simulated long dust-black carbon belt along 15N. The study indicated improvements in the results, but it is unclear how the improvements are statistically relevant. The impact of other processes could affect the dry deposition scheme but was not addressed. A statistical rather than a visual comparison between CMAQ and satellite/assimilated AOD would be more convincing to demonstrate modeling performances. Also, the manuscript contains numerous grammatical and technical errors that require thorough proofreading before resubmission.

**General Comment Response:** The authors wish to thank the reviewer for the constructive comments on our work. The present research purposely investigates how the various types of dry deposition schemes embedded within CMAQv5.4 could help improve the latest refined dust model proposed by Kong et al. (2024). The other processes that could impact the dry deposition scheme efficiency, such as resistance, land surface, roughness length …, which is not our research scope. However, we thank the reviewer for pointing out the possible research question. We will consider the idea proposed to minimize the research gap. The manuscript has been proofread through Grammarly (version Premium) online text editor

All of the changes in the revised manuscript have been highlighted in yellow. Corrections (blue text) with line numbers indicated in this response document refer to the revised manuscript.

*RC2: Specific comments and responses*

**Comment 1:** The abstract should not include references.

**Response:** The references has been removed. We modified the sentence as "By utilizing the CMAQv5.4 with the refined dust emission treatment, the East Asian dust (EAD) simulation during January 2023 was constructed to evaluate the performance of four dry deposition parameterizations, namely PR11, E20, S22, and P22." **Page 1, Line 14-17**.

**Comment 2:** L 82. What is "LABS"?

**Response:** "LABS" refers to "Lulin Atmospheric Background Station." The abbreviation has been included on **Page 1, Lines 22**.

**Comment 3:** L 115. Where is Vs in the equation?

**Response:** $V_s$ as one of the functions in the physical formulation of $V_d$. We corrected the formula as below: "

$$V_d = V_s + \frac{1}{R_a + R_s} \qquad (3)$$

where $V_s$ is the gravitation settling velocity, $R_a$ is the resistivity aerodynamic and $R_s$ is the surface resistivity. The $V_s$ is calculated according to Stokes's Law as:

$$V_s = \frac{p_p \, D_p^2 \, g C_c}{18\eta} \qquad (4)$$

where, $p_p$ is the density of the particle; $D_p$ is the diameter of the particle; g is gravitational acceleration; $C_c$ is the Cunningham correction factor for small particles; and, η is the dynamic viscosity of air." **Page 6, Line 137-142**.

**Comment 4:** L 168. The sentence is unclear. Clouds always induce biases in modeled and assimilated aerosols.

**Response:** The sentence has been revised. We modified the sentence as "The Modern Era Retrospective-analysis for Research and Application version 2 (MERRA-2) reanalysis data was used to demonstrate the spatiotemporal distribution of dust, compared with the air quality model, irrespective of the influence of clouds." **Page 8, Line 190-193.**

**Comment 5:** L 170. MERRA-2 is a data-assimilated system rather than a remotely sensed data.

**Response:** The sentence has been revised. We changed the sentence to "MERRA-2 (Gelaro et al., 2017) is a NASA reanalysis product utilizing Goddard Earth Observing System Data Assimilation System Version 5 (GEOS-5) and covering the data-assimilated system at a native spatial resolution of 0.5 ∘ × 0.625 ∘." **Page 8, Line 193-195.**

**Comment 6:** L 227. MERRA-2 is a data-assimilated product rather than a pure observational product. It's unclear how this sentence fits in with Figure 4.

**Response:** The sentence has been removed.

**Comment 7:** Fig S1. The link in the caption does not show these synoptic maps.

**Response:** The link in the caption has been revised. We corrected the caption as "Figure S2: Surface weather maps for the weather pattern obtained by Taiwan Central Weather Bureau (https://www.cwa.gov.tw/)." **Supplementary, Page 3, Line 33.**

**Comment 8** Fig S2. A statistical comparison between collocated CMAQ and MODIS AOD with a scatterplot is needed to quantify their agreements.

**Response:** We thank the reviewer for the comment. MODIS AOD retrieved consisted of the missing value due to the cloud cover. Hence, the visualized qualitive comparison between CMAQ and MODIS can be more appropriate instead of a statistical comparison. However, we agree that using a scatter plot is needed for the evaluation quantification. A detailed model evaluation between CMAQ and the observed dataset over mainland China has been delivered to carry out the statistical comparison, which is more reliable in testing the model efficiency (Table 5). We added the discussion as "Figure 5 shows the scatter plot of simulated and observed PM across mainland China. The correlation coefficient (R), a factor of two (FAC2), and the mean observed and simulated PM are marked in Figure 5. The modeled $PM_{10}$ without the dust scheme had the lowest correlation, followed by Dust_PR11. Among all of these simulations, Dust_E20 performed the best (R > 0.3) compared to Dust_PR11, Dust_S22 and Dust_P22. However, for $PM_{2.5}$, the correlation between the model and measured values was similar for all the dry deposition schemes. The statistical index of FAC2 was used in the present work since either low or high outliers less influence it (Chan and Hanna, 2004). The dataset is reliable for FAC2 values between 0.5 and 2.0, with the ideal model of 1.0. The simulated $PM_{10}$ by E20 performed well, with a nearly perfect value of 1.1. Meanwhile, the $PM_{2.5}$ by S22

simulation was slightly better than E20 but much better than the other experiments." **Page 11, Line 275-284.**

**Comment 9:** Fig S3. A statistical comparison is also needed by using the MERRA-2 AOD as well, not just the dust column. MERRA-2 provides AOD for each species.

**Response:** We thank the reviewer for the comment. Fig S5 aims to demonstrate the consistency of the transport pattern between dust and black carbon over the western Pacific Ocean, as shown by MERRA-2. Please see **Comment 8** for the detailed scatter plot analysis.

An explanation regarding the transport pattern consistency is included. We added the sentence as "Such consistency has been verified by the MERRA-2 dust and black carbon mass column over the region (red dash rectangular in Fig. S5)." **Page 15, Line 383-384.**

Also, Fig. S5 has been modified to emphasize the transboundary over the western Pacific Ocean. We change the figure as:

[Figure]

**Figure S5:** MERRA2 dust mass column (a1-a3) and black carbon mass column (b1-b3) during 06 UTC (a1, b1) 26 January, (a2, b2) 27 January and (a3, b3) 28 January 2023.

---

## Author Comment (AC3)

**Reply to the comments of RC3**

*RC3: General comment*

**General comment 1:** The manuscript aims to evaluate and compare four dry deposition schemes implemented in CMAQ v5.4, focusing on simulating an East Asian dust episode from January 2023. The authors then selected the scheme by Emerson et al. (2020) (E20) for further evaluation of dust and black carbon transport. The main issue with this work, in its current form, is that the conclusions, particularly given the strong tone used in parts of the text, are not well supported by the marginal differences between the four schemes. Specifically, the results in Table 4 and Figure 4 show small statistical significance between the models, so it's hard to justify the claim that the E20 scheme outperformed the others.

**Response:** We thank the reviewer for the comments. To re-justify the E20 scheme model performance, we have included the model evaluation over most parts of mainland China, which is identified as the dust source/near-source region. 100 observation sites have been obtained from observation sites on the Chinese air quality online monitoring analysis platform's website (www.aqistudy.cn/). The averaged observed $PM_{10}$ and $PM_{2.5}$ concentrations from the responding dataset were used to evaluate the model performance for the latest dust emission scheme and four dry deposition schemes. Also, instead of the 10-day simulation in January 2023, we widened the simulation period by considering the multiple dust storm episodes during spring 2021 for about 40 days of simulation (12 Mar-20 Apr 2021). We added the discussion as "In addition, the hourly $PM_{10}$ and $PM_{2.5}$ of nearly 100 sites distributed over mainland China (Fig. S1), covering the period of 22-31 January 2023 and 12 March-20 April 2021 obtained from the Chinese air quality online monitoring analysis platform's website (www.aqistudy.cn/)." **Page 8, Line 188-190**

Also, we have added the discussion as "During the spring of 2021, a series of dust storms (15 March, 27 March, and 18 April) occurred over the Gobi area, with one of the most significant dust storms in the past decade (15 March, the "3.15" dust storm hereafter) causing environmental impact over the continental (Jin et al., 2022; Gui et al., 2022; He et al., 2022; Liang et al., 2022; Tang et al., 2022). More interestingly, one of the multiple dust storm episodes reached western Pacific Ocean due to the extreme typhoon episode (Kong et al., 2024). Hence, we intend to re-emphasize the precision of various deposition schemes on the CMAQ for the recent dust storm episode over the Asian Continental highlighted by Kong et al. (2024). We evaluated the CMAQ simulations with the different dry deposition schemes for the 40-day sensitivity test on 12 March-20 April 2021 against measured $PM_{10}$ and $PM_{2.5}$ concentrations across the observation sites in mainland China (Table 4). The observation sites used for the model comparison are marked in Fig. S1. Generally, the evaluation results for Taiwan and mainland China were consistent. During the 40 days of Spring 2021, the CMAQ $PM_{10}$ of NMB was the highest for Off_PR11 (NMB = -79.19 %), followed by Dust_PR11 (-60.53 %). The latest inline dust emission scheme embedded with E20 dry deposition scheme for $PM_{10}$ was well performed by NMB of -25.43 %, compared to the Dust_S22 (-45.97 %) and Dust_P22 (-59.82 %). For the $PM_{2.5}$ simulation, Dust_PR11 has been improved from Dust_Off, and Dust_S22 was slightly better than Dust_E20.

Figure 5 shows the scatter plot of simulated and observed PM across mainland China. The correlation coefficient (R), a factor of two (FAC2), and the mean observed and simulated PM are marked in Figure 5. The modeled $PM_{10}$ without the dust scheme had the lowest correlation, followed by Dust_PR11. Among all of these simulations, Dust_E20 performed the

best (R > 0.3) compared to Dust_PR11, Dust_S22 and Dust_P22. However, for $PM_{2.5}$, the correlation between the model and measured values was similar for all the dry deposition schemes. The statistical index of FAC2 was used in the present work since either low or high outliers less influence it (Chan and Hanna, 2004). The dataset is reliable for FAC2 values between 0.5 and 2.0, with the ideal model of 1.0. The simulated $PM_{10}$ by E20 performed well, with a nearly perfect value of 1.1. Meanwhile, the $PM_{2.5}$ by S22 simulation was slightly better than E20 but much better than the other experiments." **Page 11, Line 259-284**.

**Table 4.** CMAQ evaluation for $PM_{10}$ and $PM_{2.5}$ against the averaged 100 observation sites across mainland China (Fig. S1) and AOD against MODIS daily observation near the dust source region (above 30°N) with Normalized Mean Bias (NMB) under the multiple simulation scenarios (Fig. S3). Spring 2021, 3.15, 3.27, and 4.18 represent the evaluation period by 12 March-20 April 2021, 14-16 March 2021, 26-28 March 2021, and 17-19 April 2021, respectively.

| Parameters | Period | CMAQ-M3DRY | | CMAQ-STAGE | | |
|---|---|---|---|---|---|---|
| | | Off_PR11 | Dust_PR11 | Dust_E20 | Dust_S22 | Dust_P22 |
| $PM_{10}$ | Spring 2021 | -79.15 | -60.53 | -25.43 | -45.97 | -59.82 |
| $PM_{2.5}$ | Spring 2021 | -60.94 | -44.84 | -37.50 | -36.29 | -42.47 |
| AOD | 3.15 | -81.92 | -49.54 | -38.97 | -46.41 | -48.45 |
| | 3.27 | -75.10 | -46.12 | -36.39 | -41.84 | -44.52 |
| | 4.18 | -55.88 | -16.49 | -3.20 | -7.83 | -14.52 |
| | Mean AOD | -70.97 | -37.38 | -26.19 | -32.03 | -35.83 |

[Figure]

**Figure 5:** The scatter plot of the observed against modeled $PM_{10}$ (a-e) and $PM_{2.5}$ (f-j) for CMAQ_Off_PR11 (a, f), CMAQ_Dust_PR11 (b, g), CMAQ_Dust_E20 (c, h), CMAQ_Dust_S22 (d, i) and CMAQ_Dust_P22 (e, j), at the 100 sites of the mainland China on 12 March-20 April 2021 (http://www.aqistudy.cn/). R is the correlation coefficient between the observation and model; FAC2 is the factor of two; MeanOBS and MeanSIM are the mean of PM from observation and model, respectively.

**General comment 2:** Additionally, results presented in various figures seem inconsistent. For instance, Table 4 suggests a clear difference between the PR11 and P22 schemes, but Figure 5 shows minimal differences between the two (see panels d & h).

**Response:** We thank the reviewer for the comments. For the reviewer's understanding, we have referred to the modeling evaluation result between STAGE and M3DRY under CMAQv5.4 conducted by USEPA (2024). According to the report, the difference in modeled particulate matter between PR11 and P22 dry deposition was within 5 %, and the time series trend was similar. To justify the present model result of PR11 and P22, we purposely replot the corresponding PM10 and PM25 for a clear comparison (Fig. S6). Moreover, the STAGE by E20 simulated a higher PM$_{2.5}$ compared to STAGE P22 (USEPA, 2024). This shows that the model testing results over the CONUS were consistent with the East Asia domain demonstrated in the present study.

[Figure]

**Figure S6:** Time series of PM$_{10}$ (left panel) and PM$_{2.5}$ (right panel) concentrations during 22-31 January 2023 under CMAQ_Dust_PR11 and CMAQ_Dust_P22 simulations over the Cape Fuguei (upper panel) and LABS (lower panel), representing the surface and high altitude, respectively.

Reference:
U.S. Environmental Protection Agency: https://github.com/USEPA/CMAQ/wiki/CMAQ-Release-Notes:-Dry-Deposition-Air-Surface-Exchange:-Surface-Tiled-Aerosol-and-Gaseous-Exchange-(STAGE), last access 15 October 2024.

**General comment 3:** Moreover, the connection between the first part of the paper, which compares four schemes, and the second part, which focuses on dust and black carbon transport, is unclear. This makes the paper feel disjointed. The same applies to the mention of the three dust storms from 2021; it's unclear how they tie into the paper's objectives.

**Response:** The connection between the first and second part of the paper has been included in the introduction section. We added the paragraphs as "The transboundary pollutants mechanisms have been widely discussed through LABS measurements, cooperating with the backward trajectory, reanalysis dataset, and modeling approach. Previous research reveals that LABS pollutants could be associated with severe fire emissions from northern Peninsular Southeast Asia (Huang et al., 2020; Ooi et al., 2021) and Indonesia (Ravindra Babu et al., 2023). Moreover, the intense wind speed in northwest China could transport the mineral dust through the surface and high-altitude layer detected at LABS (Kong et al., 2021; Kong et al., 2022). Additionally, the transport process of East Asian haze due to the cold surge from the Asian Continental industrial region towards Taiwan has been widely discussed (Chuang et al., 2020). Instead of pure aerosol, the coexistence of dust and biomass burning over Taiwan, a condition discovered in previous research, has significant implications for the regional climate (Dong et al., 2018; Dong et al., 2019). However, the high-altitude synoptic pattern associated with the coexistence between natural dust and anthropogenic pollutants remains unknown due to lack of observations at the upper layers.

This study used the chemical transport model to investigate the long-range transport of East Asian dust (EAD) that occurred on 22-31 January 2023 and 12 March-20 April 2021. Due to the limitation of the dust model, the CMAQ version 5.4, embedded with four types of dry deposition schemes, was implemented to justify the effectiveness of improving our latest refined dust model (Kong et al., 2024). LABS detected the recent transboundary episode in January 2023 as a mixing aerosol type (see Section 3.1), which has not been widely discussed, and the multiple dust storm episodes mentioned by Kong et al. (2024) provide an opportunity to model the EAD over the downwind region. Recognizing the significant transboundary events detected through Taiwan's observations, the improvement of the CMAQ dust model by the dry deposition schemes, and its application in characterizing the transport mechanism can be vital. The paper is organized as follows. The model setup and ancillary datasets are discussed in Sect. 2. The results and discussion are presented in Sect. 3, followed by the conclusions in Sect. 4." **Page 11, Line 259-284**.

**General comment 4:** Overall, the manuscript appears rushed. The figures are not properly discussed, and the text contains several grammatical and stylistic errors. It would significantly improve the paper if the authors dig deeper into the mechanistic differences between the schemes and how these differences impact the simulations.

**Response:** We thank the reviewer for the comments and suggestions. The main objectives of the research are to improve the dust model performance by using the latest deposition schemes from CMAQv5.4. This research has the potential to significantly advance our understanding of atmospheric processes. Also, the variation of the aerosol particle mode caused by the difference in dry depositions schemes was the main concern. The mechanics relating to $V_d$ and aerosol profile has been revised in Section 3.3 as "We estimated the CMAQ averaged particle modes for the PR11 dry deposition scheme and the corresponding percentage changes using E20, S22, and P22 (Fig. 8). By using E20 and S22, we found that the $V_d$ corresponding to the Aitken and accumulation modes has been increased by >100 % over most of the CMAQ domain, which was most obvious over Asian continent (Fig 8b, c, f, g). Meanwhile, the variation of $V_d$

distribution was insignificant for P22 (Fig. 8d, h, i). For the coarse mode particles, the $V_d$ has been tremendously reduced for E20 and S22 compared to PR11. However, for S22, the $V_d$ has increased by >100 % over northwest China, which is the dust source region (Fig. 8k). This leads to a significant deposition over the desert before transporting it to the downwind region, causing less $PM_{10}$ simulated by S22 than E20. A previous study proposed the $V_d$ for the aerosol at the water surface was associated with the CTM uncertainly at the downwind region (Kong et al., 2021, 2024; Ryu and Min, 2022). The $V_d$ of Aitken and accumulation modes at land and water surfaces increased generally, except E20 at the water surface. Interestingly, the coarse mode $V_d$ at the water surface for E20 and S22 decreased significantly by -44.65 % and -21.44 %, respectively, suggesting that both deposition schemes, particularly E20, could resolve the excessive deposition over the marine boundary layer (Table 5). Such minimal deposition velocity distributing over a large part of the western Pacific Ocean, including the Sea of Japan, Yellow Sea, East China Sea, and South China Sea, might be responsible for reducing the modeled $PM_{10}$ underestimation over Taiwan (Fig.8j, k), as mentioned by Kong et al. (2021)." **Page 13-14, Line 328-344**.

Also, we have revised Table 5 and Figure 8 as below:

**Table 5.** Average deposition velocity and the percentage change by PR11 corresponding to E20, S22, and P22, for Aitken, Accumulation, and Coarse modes over land and ocean boundary layer, respectively.

| Dry deposition schemes | Aitken | | Accumulation | | Coarse | |
|---|---|---|---|---|---|---|
| | Land | Ocean | Land | Ocean | Land | Ocean |
| PR11 (cm s$^{-1}$) | 0.080 | 0.062 | 0.061 | 0.042 | 0.264 | 0.109 |
| E20 (cm s$^{-1}$) | 0.090 | 0.074 | 0.065 | 0.040 | 0.139 | 0.060 |
| S22 (cm s$^{-1}$) | 0.219 | 0.117 | 0.120 | 0.064 | 0.078 | 0.085 |
| P22 (cm s$^{-1}$) | 0.085 | 0.062 | 0.072 | 0.043 | 0.290 | 0.116 |
| | | | | | | |
| $\Delta$E20 (%) | 12.66 | 20.06 | 5.43 | -5.19 | -47.10 | -44.65 |
| $\Delta$S22 (%) | 173.74 | 89.45 | 96.52 | 52.35 | -70.29 | -21.44 |
| $\Delta$P22 (%) | 6.10 | 1.37 | 17.66 | 1.52 | 10.06 | 6.86 |

Note: $\Delta$ representing the percentage change by PR11 as relative to E20, S22 and P22.

[Figure]

**Figure 8:** CMAQ estimated 10 days (22-31 January 2023) averaged for the (a-d) Aitken, (e-h) accumulation, and (i-l) coarse particle modes for PR11 dry deposition scheme (a, e, i) and the corresponding concentration percentage changes (%) using (b, f, j) E20, (c, g, k) S22 and (d, h, l) P22 schemes. Red-dash rectangular indicates the region across northwest China; Black-dash rectangular indicates the marine boundary layer.

**RC3: Specific comments and responses**

**Comment 1:** What's the significance of including Equations 1-4? They are all pretty standard and known to the community, and also were not referred to later in the text.

**Response:** We thank the reviewer for the question. Equations 1-4 represent the model algorithms used to calculate the dust emission and dry deposition. Even though the equations are not referred to the later text, the physical formulations shall be mentioned in the very beginning to introduce the calculations, which we do not treat the model a black box.

**Comment 2:** Figure 3 is barely discussed in the text.

**Response:** Figure 3 is the observation data retrieved from LABS, and has been discussed in the manuscript. We have included as "Two interesting high pollution events at Mt. Lulin (2,862 m above sea level) during 24-26 Jan and 27-30 January, respectively, are shown in Fig. 3. The latter event was more intense compared to the earlier one, where the maximum PM$_{10}$ concentration can reach up to 35 µg m$^{-3}$. Moreover, it was observed that the black carbon concentrations could reach up to a maximum of 400 ng m$^{-3}$. Based on the *in-situ* measurement, it was interesting to find the mixing state between dust, black carbon, and brown carbon (Fig.

3c). Different from what has been discussed by Kong et al. (2022), the long-range transport air pollution at the high-altitude not just merely EAD, but also included the anthropogenic pollutant from mainland China." **Page 9, Line 215-222.**

**Comment 3:** Lines 380-381: The meaning here is unclear.

**Response:** The statement has been revised. We modified the sentence as **"Additionally, the simulation of the multiple dust episodes in spring 2021 were re-constructed to evaluate the CMAQ performance over the Asian Continental. The E20 dry deposition scheme outperformed the other schemes with the lowest NMB value in simulating $PM_{10}$ (-25.4 %) and AOD (-26.2%). For the modeled $PM_{2.5}$, S22 performed slightly better than E20, with NMB of -36.29 % and -37.5 %, respectively."** **Page 17, Line 423-427**.

**Comment 4:** Lines 369-371 and 390-391: Highly subjective statements.

**Response:** The sentences have been removed.

**Comment 5:** The first part of the conclusion section repeats previously discussed results.

**Response:** We thank the reviewer for the comment. The main modeling results were repeated in conclusion to summarize the whole result and discussion.

---

## Author Response (AR2)

**Reply to the comments of RC3**

*RC3: General comment*

**General comment 1:** The revised manuscript is an improvement over the previous version. However, major issues still persist. Specifically, the objectives of the manuscript remain unclear, and the overall structure appears disjointed.

**Response:** We thank the reviewer for the comment. We apologize if our previous statement wasn't clear enough to re-join the two objectives. Based on the literature review, the dry deposition scheme by Shu et al. (2022) has reduced certain model biases, but the effect of the scheme on the East Asian Dust has not been conducted, which means the dry deposition scheme was tested without the dust scheme implementation. The same situation happened to the dry deposition scheme proposed by Emerson et al. (2020) and Pleim et al. (2022). On the contrary, Kong et al. (2024) have updated the dust emission treatment in CMAQ applied for the East Asia region, but the response towards the new dry deposition schemes needs to be tested for future application. Due to the lack of observed deposition data in the western Pacific, adequate high-altitude (LABS) and ground observation data in Taiwan are essential references for verifying the model performance. With the advantage of high-altitude observation, LABS has detected the mixing aerosol type (dust and black carbon) that has not been widely explained, proving a good opportunity for model comparison. Under all of these research gaps, the model evaluation of multiple dry deposition schemes under the recently modified dust emission treatment, with the help of Taiwan's observation, can be vital. We have re-clarified the objective as follows:

"This study used the chemical transport model to investigate the long-range transport of East Asian dust (EAD) that occurred on 22-31 January 2023 and 12 March-20 April 2021. Due to the limitation of the dust model, the CMAQ version 5.4, embedded with three types of dry deposition schemes, was implemented to justify the effectiveness of improving our latest refined dust model (Kong et al., 2024). The dry deposition scheme proposed by Shu et al. (2022) has reduced certain model bias as compared to the base scheme. However, the revised scheme response to the natural phenomenon such as wind-blown dust has not being tested. In the other way, the number of concentrations of the large size particle has been decreased over land, and increased over ocean area globally by the adjusted collective coefficiency proposed by Emerson et al. (2020). Pleim et al. (2022) has included the consideration of white cap effect which dependent on wind speed and sea surface temperature into the dry deposition scheme. Hence, the response of the CMAQ dust model under the newly developed dry deposition schemes are worth investigating in reducing the model uncertainty.

LABS detected the recent transboundary episode in January 2023 as a mixing aerosol type (see Section 3.1), which has not been widely discussed, and the multiple dust storm episodes mentioned by Kong et al. (2024) provide an opportunity to model the EAD over the downwind region. Recognizing the significant transboundary events detected through Taiwan's observations, the improvement of the CMAQ dust model by the dry deposition schemes, and its application in characterizing the transport mechanism can be vital." **Page 4-5, line 97-115 in the revised manuscript.**

**General comment 2:** The abstract fails to mention the results presented in Figures 9–12 and they are only briefly emphasized in the final paragraph of the Summary and Conclusion section. Instead, the text focuses predominantly on comparisons between the dry deposition schemes and highlights E20 as superior.

**Response:** We thank the reviewer for the comment. The abstract and conclusion included more statement of Figures 10-12 (Figures 9-12 from the previous manuscript). To examine the dynamic between both aerosol types, we modified the surface resistivity at the smooth surface ($R_b$) from P22 to see how both aerosol types behave as responses to different $R_b$. Please see **General comment 5** for more detail.

The abstract has been revised. We modified as "… On 22-31 January 2023, the *in-situ* measurement of the upper level observed the possibility of natural dust and anthropogenic aerosol. This is consistent with the CMAQ, which shows that both aerosol types displayed a clear "long dust-black carbon belt" along the 15°N. It is revealed that the increase of wet deposition due to the surface resistivity ($R_b$) leads to a significant increase in dust mass concentration but a minor increase in black carbon (BC)...." **Page 1-2, line 28-32 in revised manuscript.**

The conclusion has been accordingly revised. We modified as "…It is worth revealing that the transboundary transport of EAD from the Asian continent towards the western Pacific Ocean at the upper level was associated with the eastward moving trough system. Such transport mechanisms have been found to bring along black carbon aerosol, which is primarily the main element of China's human-made emissions. More interestingly, both aerosol profiles created a "long dust-black carbon belt" along the 15°N. The 'double dome mechanism', a concept proposed by Huang et al. (2019) that depicts the superposition of the two aerosol types, was also simulated in the present study. Besides the similarity of both, the discrepancy in the case of the aerosol deposition and mass concentration was shown. By comparing the base P22 scheme to the revised scheme (P22E01-P22E03), wet deposition increases and hence increases the dust aerosol. In other ways, black carbon aerosol also increases in a minimal magnitude, not as much as dust aerosol. This study highlights the importance of dry deposition schemes for the modeled dust and black carbon concentration and provides a reference for better dry deposition schemes in CTMs over East Asia." **Page 18-19, line 488-499 in revised manuscript.**

**General comment 3:** Combined with the marginal statistical differences observed over a relatively short simulation period—despite the addition of 40-day simulations in 2021—this weakens the manuscript's scientific impact. In its current form, the paper is neither a comprehensive model comparison study nor a detailed test case analysis. It seems to attempt to address both objectives, but unfortunately, it falls short on both fronts.

**Response:** We thank the reviewer for the comment. We agree with the reviewer that a 10-day simulation might not be sufficient for the model comparison. However, due to the critical role of LABS in the western Pacific, the model comparison under the special long-range transport case is vital. To re-justify the impact of the dry deposition scheme on the CMAQ model, the 40-days simulation (multiple dust storm episodes) was used to double-check the results from the 10-days simulation. Overall, the simulated deposition velocity ($V_d$) during the 10-day simulation (January 2023) was consistent with the 40-day simulation (Spring 2021), as shown in Figure 7. The coarse mode of $V_d$ under E20 during January 2023 and Spring 2021 was the lowest compared to S22 and P22. On the contrary, P22 recorded the highest $V_d$ during both simulations. Such consistency of $V_d$ during both periods leads to highest $PM_{10}$ concentrations by E20, and lowest $PM_{10}$ concentration by P22. The discussion on the simulated $V_d$ (S22, E20 and P22) comparing the 40-day simulation in Spring 2021 and the 10-day simulation in January

2023, has been explained in detail. We modified it as follows: "Figure 7 shows the boxplot of the averaged simulated $V_d$ for the Aitken, accumulation, and coarse particles modes under multiple deposition schemes in January 2023 (S22_2023, E20_2023, and P22_2023) and in Spring 2021 (S22_2021, E20_2021, and P22_2021). These different dry deposition treatments have a substantial impact on the aerosol profile, altering the ambient total dry deposition regionally. For instance, the median deposition velocity of S22_2023, E20_2023, and P22_2023 of the Aitken (accumulation) modes particle were 0.069 (0.020) cm s$^{-1}$, 0.039 (0.014) cm s$^{-1}$ and 0.034 (0.029) cm s$^{-1}$, respectively. The E20 simulation median $V_d$ decreased by -12.65 % for coarse-mode particles compared to S22. Also, the 75$^{th}$ percentile $V_d$ of the coarse mode has been significantly reduced by -32.13 %. On the other hand, P22 showed a different simulation by the median $V_d$ increment of 71.38 %. These findings suggest that the choice of dry deposition treatment can significantly influence the distribution and concentration of aerosols in the atmosphere, with potential implications for air quality and climate.

[revised manuscript text omitted]

**General comment 4:** Another major concern is the inconsistency between what is expected from the schemes based on their formulation and the existing literature versus what is reported in this manuscript. For example, P11 and P22 schemes are expected to be significantly different, as demonstrated in Figures 1–6 of Pleim et al. (2022). Yet, the authors report almost identical results for these models. They refer to the EPA analysis (https://github.com/USEPA/CMAQ/wiki/CMAQ-Release-Notes:-Dry-Deposition-Air-Surface-Exchange:-Surface-Tiled-Aerosol-and-Gaseous-Exchange-(STAGE)), which mentions only a 5% difference between the two schemes for CONUS. However, this analysis was not intended to compare P11 and P22 but rather to evaluate the implementation of P22 in M3Dry and STAGE. There appears to be confusion regarding the comparison of STAGE/M3Dry versus P11/P22. This can be clarified by noting that the "M3Dry" curve labeled in the EPA document (green curve) actually represents P22, not P11 (see Figures 2 or 3 of Pleim et al. (2022)).

**Response:** We thank the reviewer for the comment. The main difference between STAGE and M3DRY is the sub-grid scale variations in land use categories and the calculation of the component resistances. The present work's main objective is to compare different dry deposition schemes, namely S22, E20, and P22 on the CMAQ dust model. The present studies only focus on the STAGE dry deposition option to avoid confusion. The PR11(M3DRY) has been removed. We added the sentence as **"Since the present study is primarily focused on the impact of dry deposition scheme on CMAQ dust model, the simulations with the STAGE module are the mandatory concern." Page 7, line 162-163 in revised manuscript.**

The result and discussion have been modified, particularly Section 3.2. We modify as:

[revised manuscript text omitted]

We modified Table 1, Table 3, Table 4, Table S1, Figure 4, Figure 5, Figure 6, and Figure S3 as:

**Table 1.** Model settings.

[revised manuscript text omitted]

**General comment 5:** If the goal is to conduct a true model comparison study, I recommend that the authors look deeper into the differences between the schemes and explicitly link their case study results to the schemes' formulations.

**Response:** We agree with the reviewer for the comment. The case study, schemes and its formulation were linked for explanation. As the case study in January 2023 is essential for the model evaluation over the western Pacific, the response of P22 on dust and black carbon has been carried out. This is because the P22 dry deposition scheme in CMAQv5.4 has included the white-cap effect over the ocean surface, which is related to the particle collection efficiency by impaction as a function of $R_b$ at the smooth surface. Hence, the sensitivity of $R_b$ by P22 should be tested for the western Pacific region like Taiwan, which is surrounded by the ocean. The $R_b$ has been increased to a factor of 10, 50, and 100 under P22E01, P22E02, and P22E03, respectively. The dust aerosol can increase more significantly compared to black carbon with increased $R_b$. This means that the calibrating of $R_b$ at the smooth surface can dramatically improve the dust model, potentially revolutionizing our understanding of the impact of the dry deposition scheme on the dust model. We revised the section as follows:

[revised manuscript text omitted]

---

## Author Response (AR3)

**Reply to the comments of Editor**

**Comment 1:** Please address comments provided by the reviewer. I also have this concern: this study claims that using E20 dry deposition scheme improves model prediction of PM concentration from which it is concluded that E20 is a more accurate scheme. There are several issues related to such a statement: (1) are the differences in predicted PM concentrations between using different deposition schemes significant? I noticed that in some cases the error percentages from using different dry deposition schemes only changed marginally.

**Response:** We thank the editor for the comment. The performances of various dry deposition schemes can be significant different for $PM_{10}$, but marginally different for $PM_{2.5}$ simulation. Hence, the improvement of CMAQ by E20 dry deposition scheme was particularly for $PM_{10}$ simulation, that most represented during East Asian Dust episodes. We added the statement as "It's worth noting that E20, in particular, showed exceptional performance in the $PM_{10}$ simulation compared to other dry deposition schemes under the refined dust scheme. This underscores the potential effectiveness of E20 in managing $PM_{10}$ particulate matter. However, the $PM_{2.5}$ simulations showed only marginal changes, regardless of whether it was a surface or high-altitude simulation." **Page 11, line 282-285 in the revised manuscript.**

"Similar to the trend of $PM_{2.5}$ simulations in Taiwan (as shown in Fig. 4), the spatial distribution of the modeled $PM_{2.5}$ was identical to that of all dry deposition schemes. The result implies the significant impact of dry deposition on the EAD simulation's dust model, displaying the positive relationship between dust deposition and $PM_{10}$ concentrations (Zhang et al., 2017)." **Page 13, line 333-337 in the revised manuscript.**

We modified the statement in the abstract as "We proposed implementing the E20 dry deposition approach, particularly in $PM_{10}$ simulation, narrowing the uncertainty of the CMAQ dust emission treatment." **Page 2, line 32-34 in the revised manuscript.**

**Comment 2:** (2) Because there are compensating errors between different processes in the model, can the improved model-prediction of PM concentration be solely attributed to the chosen dry deposition scheme? For example, if dust emission is biased low, then using a deposition scheme with low Vd would give a better prediction of PM concentration, and vice versa. This does not mean such a deposition scheme is more accurate than others. In other words, can the conclusion presented here be generalized? I would recommend the authors to either add more uncertainty analysis/discussion to support their conclusion and/or rephase their conclusion by considering the factors mentioned above.

**Response:** We appreciate the editor's constructive comments and suggestions. We agree with the editor that the performance of dust emission could influence the PM prediction with a certain level of $V_d$. The higher dust emission (or considered as low bias) leads to lower $V_d$ and, eventually, higher dust loading. The present work intentionally simulated the strong dust intensity of multiple dust storms in Spring 2021 and a regular EAD episode in January 2023. The purpose is to investigate whether or not the distinct dust emission intensity could impact aerosol deposition and concentrations. The result shows that the spring 2021 simulation generated higher dust emissions than January 2023, which caused lower $V_d$ during Spring 2021 compared to the one during January 2023 (Figure 7). Under both strong and regular dust emissions phenomena, E20 seems to outperform the $PM_{10}$ simulation. We modified the discussion/analysis as "As shown in Figure 7a, the results during the spring of 2021 are similar

to those for January 2023 in comparing the dry deposition schemes. Notably, the $V_d$ of the coarse mode for E20_2023 and E20_2021 was lowest compared to the other dry deposition schemes. Contrary, the accumulation and coarse mode by P22 were the highest. The result was consistent with the best simulated $PM_{10}$ by E20 in 2023 and 2021 displayed in Table 4 and 5, respectively. The lowest $V_d$ of the coarse mode particle was responsible for reducing the $PM_{10}$ simulation underestimation, consistent with the simulation by Ryu and Min (2022). The slow $V_d$ means the total loss of aerosol to the surface has been minimized, leading to increased aerosol concentration. In addition, the spatial distribution of dust emissions could significantly influence the aerosol deposition velocity. The total dust emission in Spring 2021 was of a much higher magnitude and wider spatial distribution than in January 2023 (Fig. 7b, c). This led to a slow $V_d$ in the coarse mode, particularly, causing more dust loading during the multiple dust storms in Spring 2021 than the regular dust episode recorded in January 2023. This finding is consistent with Zeng et al. (2020), which emphasized the sensitivity of different dust emissions on dry deposition schemes. However, it's important to note that the research was only conducted in one particular short period. On the other hand, this work has highlighted the distinct dust emission according to EAD intensity impacting the various dry deposition schemes. These implications are crucial for understanding the behaviour of aerosols in the atmosphere and their significant impact on air quality." **Page 14, line 350-366 in the revised manuscript.**

[Figure]

**Figure 7:** (a) 10-days (2023) and 40-days (2021) averaged dry $V_d$ predicted by CMAQ for the Aitken, accumulation, and coarse particle modes using the 2023_S22 (red), 2023_E20 (orange), 2023_P22 (blue), 2021_S22 (violet), 2021_E20 (yellow) and 2021_P22 (green) particle dry deposition schemes. The variability illustrated by the boxes and whiskers corresponds to spatial variability in annually averaged values throughout the CMAQ domain. The simulated total dust emission by CMAQ_Dust_E20 in (b) January 2023 and (c) Spring 2021.

Also, the present research intends to test the dry deposition scheme's response to the dust emission model as constant variables. Even though we simulated the two different dust emission intensities, the different dust emission schemes' reactions to the dry deposition scheme in the CMAQ model remain uncertain. As a result, we propose the potential future studies, which will explore the sensitivity of the various dust emission parameterizations impacting the dry deposition schemes. We modified the conclusion as "We noted that the improved model simulation for EAD relied on dust emission, dust deposition, and transport processes. The dust emission treatment was proven sensitive to the CMAQ model performance in East Asia (Dong et al., 2018; Liu et al., 2021; Kong et al., 2024). In addition, the CTM

performance can be attributed to the dust emission schemes and the dry deposition schemes (Zeng et al., 2020). In other words, different dust emission schemes may impact the $V_d$ and dust loading, which reacts differently to model performance. The present research, which is a complex examination, is of significant importance as it primarily focuses on which dry deposition scheme can improve the most recent updated dust emission model. Therefore, the sensitivity of the dust emission parameterizations or approaches, including surface roughness, land surface, soil texture, and types on the dry deposition scheme, underscores the need for a comprehensive understanding and is proposed for future studies.

Finally, it is necessary to point out that the dry deposition on the EAD is closely associated with the $PM_{10}$ concentration (Zhang et al., 2017). Nevertheless, it has been shown that there are other atmospheric processes related to the air quality over the Western Pacific, including transboundary haze, biomass burning, and local emission (Chuang et al., 2020; Ooi et al., 2021; Chang et al., 2022). These complex phenomena could cause variations of $PM_{2.5}$, ozone, and the corresponding primary pollutant. Hence, the role and response of the dry deposition scheme in the CMAQ should be paid attention to in the future for compressive understanding and model improvement. This research enhances our understanding of dust emission and dry deposition models and provides valuable insights for improving air quality models, which is crucial for environmental and public health management." **Page 20, line 515-534 in the revised manuscript.**

**Reply to the comments of RC4**

**Comment 1:** Abstract:"The result showed that the dry deposition parameterization could significantly impact the CMAQ dust emission treatment". It seems that dust emission has no direct relationship with dry deposition parameterization. So this sentence can be replaced by "The result showed that the dry deposition parameterization could significantly impact the CMAQ dust concentration in the air"

**Response:** We thank the reviewer for the suggestion. The sentence has been revised as suggested. We modified as "The result showed that the dry deposition parameterization could significantly impact the CMAQ dust concentration in the air." **Page 1, line 17-18 in the revised manuscript.**

**Comment 2:** "It is revealed that the increase of wet deposition due to the surface resistivity (Rb) leads to a significant increase in dust mass concentration but a minor increase in black carbon (BC)." The sentence can not be understood clearly. Here, wet deposition was related to the surface resistivity (Rb)?

**Response:** We thank the reviewer for pointing out the problem. The statement specifically emphasized the significant influence of surface resistivity on the dust and minor on black carbon concentrations. We revised the sentence as "It is revealed that the increase of the surface resistivity ($R_b$) leads to a significant increase in dust mass concentration but a minor increase in black carbon (BC)." **Page 2, line 31-32 in the revised manuscript.**

Also, we have modified the conclusion. We revised the sentence as "By comparing the base P22 scheme to the revised scheme (P22E01-P22E03), the dust aerosol increased significantly and marginally by the black carbon." **Page 20, line 510-511 in the revised manuscript.**

**Comment 3:** P 148 "Vs" seems to be "Vg"

**Response:** We thank the reviewer for the comment. The symbol has been corrected. We changed the symbol as "$V_g$". **Page 6, line 148 in the revised manuscript.**

**Comment 4:** P162 "Since the present study is primary focused on the impact of dry deposition scheme on CMAQ dust model" here, "model" should be "modeling"

**Response:** We thank the reviewer for the warm reminder. The term has been revised. We corrected as "…modeling…". **Page 7, line 163 in the revised manuscript.**

**Comment 5:** P187-193, E22 and P22 refers to dry deposition of Emerson et al. (2020) and Pleim et al. (2022) . Why S22 stands for Shu et al. (2011) scheme? The author stated that "Indeed, both CMAQ_Off_S22 and CMAQ_Dust_S22 used the dry deposition mechanism by Shu et al. (2011)." I can not find the paper of Shu et al. (2011) in the reference list. Shu et al.(2022) and Pleim et al. (2022) are also missing.

**Response:** We thank the reviewer for the warm reminder. The reference Shu et al. (2011) has been removed, and replaced by Shu et al. (2022). The related references have been included to the reference list. We added the references as

"Pleim, J. E., Ran, L., Saylor, R. D., Willison, J., and Binkowski, F. S.: A New Aerosol Dry Deposition Model for Air Quality and Climate Modeling, J. Adv. Model. Earth Syst., 14, 1–21, https://doi.org/10.1029/2022MS003050, 2022." **Page 26, line 678-680 in the revised manuscript.**

"Shu, Q., Murphy, B., Schwede, D., Henderson, B. H., Pye, H. O. T., Appel, K. W., Khan, T. R., and 534 Perlinger, J. A.: Improving the particle dry deposition scheme in the CMAQ photochemical modeling 535 system, Atmos. Environ., 289, 119343, https://doi.org/10.1016/j.atmosenv.2022.119343, 2022." **Page 26, line 695-697 in the revised manuscript.**

Also, the sentence has been revised. We corrected the sentence as "Indeed, both CMAQ_Off_S22 and CMAQ_Dust_S22 used the dry deposition mechanism by Shu et al. (2022)." **Page 8, line 191-192 in the revised manuscript.**

**Comment 6:** It was suggested that the author should give more detailed description and main difference about the three dry deposition schemes of S22, E22 and P22, which can be included in a Table.

**Response:** We thank the reviewer for the constructive suggestion. We have included the detailed description of the 3 dry deposition schemes of S22, E20 and P22 in Table 1. We added as

"**Table 1.** Detailed mechanism expression relating the three dry deposition schemes.

| Schemes | Surfaces | S22 (CMAQv5.3 and beyond) | E20 | P22 |
|---|---|---|---|---|
| $V_d$ | | $f_{veg}V_{d\ vegetated} + (1-f_{veg})V_{d\ smooth}$ | $f_{veg}V_{d\ vegetated} + (1-f_{veg})V_{d\ smooth}$ | $f_{veg}V_{d\ vegetated} + (1-f_{veg})V_{d\ smooth}$ |
| $R_b$ | Vegetated | $\dfrac{1}{f_{veg}((\max{(LAI,1.0)})u_*(E_B + E_{Im}))}$ | $\dfrac{1}{wet*E_{Tot\ veg} + (1-wet)*E_{Tot\ veg}*R1}$ | $\dfrac{1}{f_{veg}((\max{(LAI,1.0)})u_*(E_B + E_{Im}))}$ |
| $R_b$ | Smooth | $\dfrac{1}{u_*(E_B + E_{Im})}$ | $\dfrac{1}{wet*E_{Tot\ smth} + (1-wet)*E_{Tot\ smth}*R1}$ | $\dfrac{1}{BAI.u_*(E_B + E_{Im})}$ |
| $E_B$ | Vegetated | $Sc^{-2/3}$ | $C_B Sc^{-2/3}$ | $C_{IB} Sc^{-2/3}$ |
| $E_B$ | Smooth | $Sc^{-2/3}$ | $C_B Sc^{-2/3}$ | $f_{wc}\dfrac{u_*}{U_{10}} + (1-f_{wc})C_{IB} Sc^{-2/3}$ |
| $E_{Im}$ | Vegetated | $\dfrac{St^2}{St^2 + 1}$ | $C_{Im}(\dfrac{St}{St+\alpha})^{1.7}$ | $f_{micro}\dfrac{Sth^2}{Sth^2+1} + (1-f_{micro})\dfrac{St1^2}{St1^2+1}$ |
| $E_{Im}$ | Smooth | $\dfrac{St^2}{St^2 + 400}$ | $C_{Im}(\dfrac{St}{St+100})^{1.7}$ | $10^{-3/St}$ |
| $E_{In}$ | Vegetated | $0$ | $C_{In}\left(\dfrac{d_p}{A}\right)^{0.8}$ | $0$ |
| $E_{In}$ | Smooth | $0$ | $0$ | $0$ |

$V_{d\ vegetated}$ = deposition velocity over the vegetative surface: $V_{d\ vegetated} = \dfrac{V_g}{1-\exp{(-V_g(R_a + R_{b\ vegetated}))}}$

$V_{d\ smooth}$ = deposition velocity over the smooth surface: $V_{d\ smooth} = \dfrac{V_g}{1-\exp{(-V_g(R_a + R_{b\ smooth}))}}$

$f_{veg}$ = grid scale vegetation-coverage fraction
$E_B$ = Brownian diffusion efficiency
$E_{Im}$ = Impaction efficiency
$E_{In}$ = Interception efficiency
Sc = Schmidt number
$St$ = Stokes number
wet = Wet surface
$E_{Tot\ veg}$ = veg_ustar*$(E_B + E_{Im} + E_{In})$
$E_{Tot\ smth}$ = 3.0*ustg*$(E_B + E_{Im})$
R1 = Bounce correction term by Slinn (1982).
$C_B$ = Brownian collective coefficient: 0.2
$C_{Im}$ = Impaction collective coefficient: 0.4
$C_{In}$ = Interception collective coefficient: 2.5
α = Empirical constant
LAI = Leaf area index
BAI = Building area index
$C_{IB}$ = 1.0/3.0
$f_{wc}$ = Whitecap surface fraction
$f_{micro}$ = Total impaction fraction from the microscale features
$u_*$ and $U_{10}$ = Frictional velocity and wind speed at 10 m (ms$^{-1}$)

$St1$ and $Sth$ = Obstacle characteristic dimensions for the leaf hairs and microscale roughness on leaves"

**Comment 7:** P187 "the present research conducted five simulation scenarios, namely CMAQ_Off_S22, CMAQ_Dust_S22, CMAQ_Dust_E20 and CMAQ_Dust_P22", here, five or four?

**Response:** We thank the reviewer for the comment. We revised the sentence as "To ensure the precision of the multiple dry deposition parameterizations, the present research conducted four simulation scenarios, namely CMAQ_Off_S22, CMAQ_Dust_S22, CMAQ_Dust_E20 and CMAQ_Dust_P22." **Page 8, line 186-188 in the revised manuscript.**